# Robust Sequence Submodular Maximization

Gamal Sallam[1], Zizhan Zheng[2], Jie Wu[1], and Bo Ji [1, 3]

[1]Department of Computer and Information Sciences, Temple University
[2]Department of Computer Science, Tulane University
[3]Department of Computer Science, Virginia Tech

## Abstract

Submodularity is an important property of set functions and has been extensively studied in the literature. It models set functions that exhibit a diminishing returns property, where the marginal value of adding an element to a set decreases as the set expands. This notion has been generalized to considering sequence functions, where the order of adding elements plays a crucial role and determines the function value; the generalized notion is called sequence (or string) submodularity. In this paper, we study a new problem of robust sequence submodular maximization with cardinality constraints. The robustness is against the removal of a subset of elements in the selected sequence (e.g., due to malfunctions or adversarial attacks). Compared to robust submodular maximization for set function, new challenges arise when sequence functions are concerned. Specifically, there are multiple definitions of submodularity for sequence functions, which exhibit subtle yet critical differences. Another challenge comes from two directions of monotonicity: forward monotonicity and backward monotonicity, both of which are important to proving performance guarantees. To address these unique challenges, we design two robust greedy algorithms: while one algorithm achieves a constant approximation ratio but is robust only against the removal of a subset of contiguous elements, the other is robust against the removal of an arbitrary subset of the selected elements but requires a stronger assumption and achieves an approximation ratio that depends on the number of the removed elements. Finally, we generalize the analyses to considering sequence functions under weaker assumptions based on approximate versions of sequence submodularity and backward monotonicity.

## 1  Introduction

Submodularity is an important property of set functions and has been extensively studied in the literature [1–4]. It models set functions that exhibit a diminishing returns property, where the marginal value of adding an element to a set decreases as the set expands (i.e., contains more elements). The notion of submodularity has been generalized to considering sequence functions, where the order of adding elements plays a crucial role and determines the function value; the generalized notion is called sequence (or string) submodularity [5–11]. Several real-world applications, including machine learning based recommendation systems, ads allocation, and automation and control, involve the selection of elements in sequence. In this paper, we study a new problem of robust sequence submodular maximization with cardinality constraints. The robustness is against the removal of a subset of elements in the selected sequence (e.g., due to malfunctions or adversarial attacks). To motivate the new problem studied in this paper, we begin with the discussions about two concrete applications (sensor activation and movie recommendation) and use them to illustrate the key differences between set functions and sequence functions.

Table 1: Representative work on submodular maximization

|  | (Set) Submodular Maximization | Sequence Submodular Maximization |
|---|---|---|
| Non-robust | [1–4] | [5–11] |
| Robust | [13–25] | **This paper** |

**Sensor Activation**. Consider the problem of sensor activation for moving target detection [5], where the objective is to sequentially activate a certain number of sensors to maximize the probability of detecting a moving target. Suppose that each sensor covers a certain area. Without any prior knowledge of the target location or the probability of detection at each location, it is plausible to maximize the total area covered by the activated sensors. If the coverage area of each sensor remains constant over time, then it is sufficient to decide which subset of sensors to activate without concerning the order in which the sensors are activated. This scenario can typically be modeled as maximizing a (set) submodular function.

In practice, however, the coverage area of each sensor may decay over time for several reasons, such as battery decay and corrosive environment. Accounting for such factors, the coverage area of a sensor may be modeled as a decreasing function, e.g., in the form of $Ce^{-t/T}$, where $C$ is the initial coverage area, $T$ is the sensor's lifetime, and $t = 0, 1, \ldots$ is the time index. In this scenario, the sequence in which the sensors are activated is of critical importance. The activation sequence determines the total coverage area and thus impacts the probability of successful detection.

The above problem becomes more challenging if some sensors may be malfunctioning after they are activated or, even worse, if there is an adversary that may attack some sensors and render them non-working. Depending on how critical the application scenario is, one must ensure resilience of the sensor activation plan such that a certain performance (i.e., probability of successful detection) can still be guaranteed even in the worst-case failure scenario.

**Movie Recommendation**. Consider that a service provider (e.g., Netflix) would like to recommend movies to a user [6, 12]. It is common that the service provider recommends movies of similar flavor to a user. To some user, however, the incremental level of entertainment of watching a movie may decrease if the user had watched more similar movies, which exhibits a diminishing returns property.

In reality, however, the order in which the movies are recommended to the user may also impact how the user perceives a specific movie. In fact, movie recommendation and TV show recommendation have been modeled as sequence functions in [6] and [12], respectively. As noted in the motivating example in [12], if the model determines that the user might be interested in *The Lord of the Rings* series, then recommending *The Return of the King* first and *The Fellowship of the Ring* last could make the user unsatisfied with an otherwise excellent recommendation. Moreover, the user may not watch all the recommended videos possibly because the user has already watched some of them or does not like them (e.g., due to low ratings and/or unfavorable reviews).

While the problem of submodular maximization is generally NP-hard due to its combinatorial nature, the property of submodularity has been exploited to design efficient approximation algorithms. Since the seminal work in [1], it has been well known that a simple greedy algorithm and its variants can achieve an optimal approximation ratio[1] of $(1 - 1/e)$ in various settings. Specifically, a variant of the greedy algorithm has also been shown to achieve the same approximation ratio for the problem of sequence submodular maximization [7]. Recently, robust versions of the submodular maximization problem have aroused a lot of research interests (e.g., [13, 15, 16]). The focus of these studies is on selecting a set of elements that is robust against the removal of a subset of them.

In this paper, we take one step further and consider a robust version of the sequence submodular maximization problem. The goal is to select a sequence of elements (i.e., elements in a specific order) with cardinality constraints such that the value of the sequence function is maximized when a certain number of the selected elements may be removed. In Table 1, we position our work along with the literature on submodular maximization. The generalization to robust sequence submodular maximization introduces new challenges. As far as sequence functions are concerned, not only do the notions of submodularity and monotonicity involve variants of subtle yet critical differences, but

the design and analysis of robust algorithms are also faced with novel technical difficulties, which render the proofs more challenging. In the sequel, we elaborate on these unique challenges.

First, there are two definitions of submodularity for set functions: (i) the marginal value of adding an element to a set decreases as the set expands; (ii) the marginal value of adding a set to another set decreases as the latter set expands. It is trivial to show that these two definitions are equivalent. Replacing "set" with "sequence" in the above definitions gives two similar definitions for sequence functions. We will show that while (ii) still trivially implies (i) for sequence functions, the opposite does not hold in general. This leads to the following important question: Which definition of submodularity should one consider for sequence functions? Interestingly, while the weaker form (i) is sufficient for establishing provable approximation guarantees for non-robust sequence submodular maximization, one needs the stronger form (ii) to obtain similar results for the robust counterpart.

Second, while monotonicity is a straightforward notion for set functions (i.e., the value of a set function does not decrease as the set expands), there are two directions of monotonicity for sequence functions: forward monotonicity and backward monotonicity. Forward (resp., backward) monotonicity means that adding a sequence to the end (resp., beginning) of another sequence does not decrease the overall value. Both monotonicity properties are important to proving performance guarantees.

Third, the impact of removing an element from a sequence depends both on the element itself and on its position in the sequence. This makes the robust algorithms designed for set functions inapplicable here and calls for new robust algorithms that are better suited for sequence functions. Besides, one needs stronger assumptions to proceed with performance analysis for sequence functions. Therefore, it is more important to prove performance guarantees under weaker assumptions based on approximate versions of submodularity and monotonicity, which are more likely to hold in practice.

Due to these unique challenges, it is unclear what conditions are sufficient for establishing provable approximation ratios for robust sequence submodular maximization, how to design efficient and robust algorithms, and how to prove performance guarantees for the designed algorithms. We aim to answer these questions in this paper. Our contributions are summarized as follows.

- To the best of our knowledge, this is the first work that considers the problem of robust sequence submodular maximization. It is well known that the traditional (set) submodular maximization problem is already NP-hard. Accounting for sequence functions and robustness guarantees adds extra layers of difficulty, as the submodular and monotone properties of sequence functions involve variants with subtle yet critical differences.

- To address these unique challenges, we design two robust greedy algorithms for maximizing a forward-monotone, backward-monotone, and sequence-submodular function with cardinality constraints. While one algorithm achieves a constant approximation ratio but is robust only against the removal of a subset of contiguous elements, the other is robust against the removal of an arbitrary subset of the selected elements but requires a stronger assumption and achieves an approximation ratio that depends on the number of the removed elements. Although our proposed greedy algorithms are quite intuitive, the theoretical analysis is more challenging, and the presented approximation guarantees are highly nontrivial.

- We consider different definitions of submodularity and monotonicity and investigate their impacts on the derived theoretical results. Our study reveals that compared to set functions, one needs more properties of sequence functions to establish similar approximation results. On the other hand, we introduce general versions of such properties, such as approximate sequence submodularity and approximate backward monotonicity, and leverage them to prove approximation results of the proposed algorithms under weaker assumptions, which are more likely to hold in practice. We hope that this work serves as an important first step towards the design and analysis of efficient algorithms for robust sequence submodular maximization, which is worth further investigation through empirical evaluations for specific applications.

Due to space limitations, we provide all the proofs in the supplementary document.

## 2 System Model and Problem Formulation

Consider a set of elements $\mathcal{V}$, with $V = |\mathcal{V}|$, where $|\cdot|$ denotes the cardinality of a set. Let $(v_1, \ldots, v_m)$ be a sequence of non-repeated[2] elements selected over $m$ steps, where $v_i \in \mathcal{V}$ for $i = 1, \ldots, m$, $v_i \neq v_j$ for $i \neq j$, and $m = 0, \ldots, |\mathcal{V}|$. When $m = 0$, the sequence is empty and is denoted by $()$. We use $\mathcal{H}(\mathcal{V})$ to denote the set of all possible sequences of non-repeated elements in $\mathcal{V}$, and we use $\mathcal{V}(S)$ to denote the set of elements in sequence $S \in \mathcal{H}(\mathcal{V})$. By slightly abusing the notation, we use $|S|$ to denote the number of elements in sequence $S$, i.e., $|S| = |\mathcal{V}(S)|$. Consider a sequence $S \in \mathcal{H}(\mathcal{V})$ and a set $\mathcal{U} \subseteq \mathcal{V}$. We use $S - \mathcal{U}$ to denote the sequence that is constructed by removing all the elements in $\mathcal{U}$ from sequence $S$ without changing the order of the remaining elements. For instance, suppose $S = (v_2, v_1, v_5, v_3)$ and $\mathcal{U} = \{v_2, v_4, v_5\}$. Then, we have $S - \mathcal{U} = (v_1, v_3)$. For two sequences $S_1, S_2 \in \mathcal{H}(\mathcal{V})$, sequence $S_1$ is said to be a subsequence of sequence $S_2$ if we can write $S_1$ as $S_2 - \mathcal{U}$ for some $\mathcal{U} \subseteq \mathcal{V}(S_2)$. Consider two sequences $S_1 = (v_1, \ldots, v_{m_1})$ and $S_2 = (u_1, \ldots, u_{m_2})$ in $\mathcal{H}(\mathcal{V})$, and let $S_2 - \mathcal{V}(S_1) = (w_1, \ldots, w_{m_3})$. We define a concatenation of $S_1$ and $S_2$ as

$$S_1 \oplus S_2 \triangleq (v_1, \ldots, v_{m_1}, w_1, \ldots, w_{m_3}). \tag{1}$$

Note that the concatenated sequence $S_1 \oplus S_2$ has no repeated elements. We write $S_1 \preceq S_2$ if we can write $S_2$ as $S_1 \oplus S_3$ for some $S_3 \in \mathcal{H}(\mathcal{V})$.

Before we define the problem of sequence submodular maximization, which was first considered in [7], we introduce some important definitions. Consider a sequence function $h : \mathcal{H}(\mathcal{V}) \to \mathbb{R}^+$, where $\mathbb{R}^+$ is the set of non-negative real numbers. Without loss of generality, we assume that the value of an empty sequence is zero, i.e., $h(()) = 0$. We define the marginal value of appending sequence $S_2$ to sequence $S_1$ as $h(S_2|S_1) \triangleq h(S_1 \oplus S_2) - h(S_1)$. Function $h$ is said to be sequence-submodular if for all $S_3 \in \mathcal{H}(\mathcal{V})$, we have

$$h(S_3|S_1) \geq h(S_3|S_2), \ \forall S_1, S_2 \in \mathcal{H}(\mathcal{V}) \text{ such that } S_1 \preceq S_2. \tag{2}$$

The above inequality represents a diminishing returns property. Similarly, function $h$ is said to be element-sequence-submodular if for all $v \in \mathcal{V}$, we have

$$h((v)|S_1) \geq h((v)|S_2), \ \forall S_1, S_2 \in \mathcal{H}(\mathcal{V}) \text{ such that } S_1 \preceq S_2. \tag{3}$$

From the above definitons, it is easy to see that a sequence-submodular function must also be element-sequence-submodular. However, an element-sequence-submodular function may not necessarily be sequence-submodular; we provide such an example in our supplementary material. This is in contrary to submodular (set) functions, for which one can easily verify that similar definitions of Eqs. (2) and (3) imply each other. Although it is noted (without a proof) in [5] that using an induction argument, one can show that an element-sequence-submodular function must also be sequence-submodular, we find this claim false due to the counterexample we find (see our supplementary material).

Also, function $h$ is said to be forward-monotone if

$$h(S_1 \oplus S_2) \geq h(S_1), \ \forall S_1, S_2 \in \mathcal{H}(\mathcal{V}), \tag{4}$$

and is said to be backward-monotone if

$$h(S_1 \oplus S_2) \geq h(S_2), \ \forall S_1, S_2 \in \mathcal{H}(\mathcal{V}). \tag{5}$$

For the sensor activation example we discussed in the introduction, forward monotonicity (resp., backward monotonicity) means that adding a sequence of sensors to the end (resp., the beginning) of another sequence of sensors does not reduce the total coverage area. We will later introduce approximate versions of sequence submodularity and backward monotonicity and generalize the theoretical results under weaker assumptions based on such generalized properties (see Section 4).

The problem of selecting a sequence $S \in \mathcal{H}(\mathcal{V})$ with an objective of maximizing function $h$ with cardinality constraints (i.e., selecting no more than $k$ elements for $k > 0$) can be formulated as

$$\max_{S \in \mathcal{H}(\mathcal{V}), \ |S| \leq k} h(S). \tag{P}$$

Next, we propose a robust version of Problem $(P)$, which accounts for the removal of some of the selected elements. Consider $\tau \leq k$. The robust version of Problem $(P)$ can be formulated as

$$\max_{S \in \mathcal{H}(\mathcal{V}),\ |S| \leq k} \min_{\mathcal{V}' \subseteq \mathcal{V}(S),\ |\mathcal{V}'| \leq \tau} h(S - \mathcal{V}'). \tag{R}$$

Without loss of generality, we assume $k > 1$ for Problem $(R)$. In the next section, we discuss the challenges of Problem $(R)$ and present the proposed robust algorithms.

## 3 Proposed Robust Algorithms

We begin with a discussion about the non-robust sequence submodular maximization problem (Problem $(P)$), through which we provide useful insights into the understanding of the challenges of Problem $(R)$. Although Problem $(P)$ is NP-hard, it can be approximately solved using a simple *Sequence Submodular Greedy* (SSG) algorithm [7]. Under the SSG algorithm, we begin with an empty sequence $S$; in each iteration, we choose an element that leads to the largest marginal value with respect to $S$ and append it to sequence $S$, i.e., $S = S \oplus \arg\max_{v \in \mathcal{V} \setminus \mathcal{V}(S)} h((v)|S)$. We repeat the above procedure until $k$ elements have been selected. It has been shown in [7] that the SSG algorithm achieves an approximation ratio of $(1 - 1/e)$ for maximizing a forward-monotone, backward-monotone, and element-sequence-submodular function with cardinality constraints.

Although the SSG algorithm approximately solves Problem $(P)$, it can perform very poorly if one directly applies it to solving its robust counterpart (Problem $(R)$). The intuition behind this is the following. The SSG algorithm tends to concentrate the value of the selected sequence on the first few elements. Selecting elements in this manner leaves the overall sequence vulnerable as removing some of these elements would have a high impact on the overall value. Consider the following example, where we assume $\tau = 1$ for simplicity. Let $\mathcal{V}_1 = \{v\}$, $\mathcal{V}_2 = \{u_1, \ldots, u_n\}$, $\mathcal{V}_3 = \{w_1, \ldots, w_n\}$, and $\mathcal{V} = \mathcal{V}_1 \cup \mathcal{V}_2 \cup \mathcal{V}_3$. Assume $h((v)) = 1$, $h((u_i)) = 1/n$ for all $u_i \in \mathcal{V}_2$, and $h((w_i)) = \epsilon$ for all $w_i \in \mathcal{V}_3$, where $\epsilon$ is an arbitrarily small positive number. Also, assume that for any $S_1, S_2 \in \mathcal{H}(\mathcal{V})$ such that $v \in \mathcal{V}(S_1)$ and $v \notin \mathcal{V}(S_2)$, we have $h((u_i)|S_1) = 0$ and $h((u_i)|S_2) = 1/n$ for all $u_i \in \mathcal{V}_2$ and $h((w_i)|S_1) = h((w_i)|S_2) = \epsilon$ for all $w_i \in \mathcal{V}_3$. Suppose $k = n$. Then, the SSG algorithm will select $v$ as the first element and select the subsequent $n - 1$ elements from $\mathcal{V}_3$. The value of the selected sequence will be $1 + (n - 1)\epsilon$. If element $v$ is removed, then the value of the remaining sequence will be $(n - 1)\epsilon$, which can be arbitrarily small. In contrast, a sequence consisting of $n$ elements from $\mathcal{V}_2$ will be robust against the removal of any element. This is because the overall value is equally distributed across all the elements in the sequence and the value of the sequence after removing any element is $(n - 1)/n$.

The above example shows that the SSG algorithm may perform arbitrarily bad for Problem $(R)$. To that end, we propose two greedy algorithms that can address this limitation and ensure robustness of the selected sequence for Problem $(R)$. First, we propose an algorithm that achieves a constant approximation ratio but is robust only against the removal of $\tau$ contiguous elements (Section 3.1). Then, we further propose an algorithm that works in a general setting without the contiguous restriction and is robust against the removal of an arbitrary subset of $\tau$ selected elements, but it requires a stronger assumption and achieves an approximation ratio that depends on the number of removed elements (Section 3.2).

### 3.1 Robustness Against the Removal of Contiguous Elements

In this subsection, we wish to design an algorithm that is robust against the removal of $\tau$ contiguous elements. The assumption of the removal of contiguous elements can model a spatial relationship such as sensors in close proximity or a temporal relationship such as consecutive episodes of a TV show. We design a variant of the SSG algorithm that approximately solves Problem $(R)$. The algorithm is presented in Algorithm 1. As we discussed earlier, the limitation of the SSG algorithm is that the selected sequence is vulnerable because the overall value might be concentrated in the first few elements. Algorithm 1 is motivated by this key observation and works in two steps. In Step 1, we select a sequence $S_1$ of $\tau$ elements from elements in $\mathcal{V}$ in a greedy manner as in SSG. In Step 2, we select another sequence $S_2$ of $k - \tau$ elements from elements in $\mathcal{V} \setminus \mathcal{V}(S_1)$, again in a greedy manner as in SSG. Note that when we select sequence $S_2$, we perform the greedy selection as if sequence $S_1$ does not exist at all. This ensures that the value of the final sequence $S = S_1 \oplus S_2$ is

| **Algorithm 1** Robust greedy algorithm against the removal of contiguous elements | **Algorithm 2** Robust greedy algorithm against the removal of arbitrary elements |
|---|---|
| 1: Input: $\mathcal{V}, k, \tau$; Output: $S$ | 1: Input: $\mathcal{V}, k, \tau$; Output: $S$ |
| 2: Initialization: $S = S_1 = S_2 = ()$ | 2: Initialization: $S = S_1 = S_2 = ()$ |
| //**Step 1:** | //**Step 1:** |
| 3: **while** $|S_1| < \tau$ **do** | 3: **while** $|S_1| < \tau$ **do** |
| 4: $\quad S_1 = S_1 \oplus \arg\max_{v \in \mathcal{V} \setminus \mathcal{V}(S_1)} h((v)|S_1)$ | 4: $\quad S_1 = S_1 \oplus \arg\max_{v \in \mathcal{V} \setminus \mathcal{V}(S_1)} h((v))$ |
| 5: **end while** | 5: **end while** |
| //**Step 2:** | //**Step 2:** |
| 6: **while** $|S_2| < k - \tau$ **do** | 6: **while** $|S_2| < k - \tau$ **do** |
| 7: $\quad S_2 =$ $\quad S_2 \oplus \arg\max_{v \in \mathcal{V} \setminus (\mathcal{V}(S_1) \cup \mathcal{V}(S_2))} h((v)|S_2)$ | 7: $\quad S_2 =$ $\quad S_2 \oplus \arg\max_{v \in \mathcal{V} \setminus (\mathcal{V}(S_1) \cup \mathcal{V}(S_2))} h((v)|S_2)$ |
| 8: **end while** | 8: **end while** |
| 9: $S = S_1 \oplus S_2$ | 9: $S = S_1 \oplus S_2$ |

not concentrated in either $S_1$ or $S_2$. The complexity of Algorithm 1 is $O(kV)$, which is in terms of the number of function evaluations used in the algorithm.

We first state the following assumption that is needed for deriving the main results in this subsection.

**Assumption 1.** *Function $h$ is forward-monotone, backward-monotone, and sequence-submodular.*

In Theorem 1, we state the approximation result of Algorithm 1 in a special case of $\tau = 1$. We consider this special case for two reasons: (i) it is easier to explain the key ideas in the proof of this special case; (ii) we can prove better approximation ratios in this special case, which may not be obtained from the analysis in the case of $1 \leq \tau \leq k$.

**Theorem 1.** *Consider $\tau = 1$. Under Assumption 1, Algorithm 1 achieves an approximation ratio of* $\max\left\{\frac{e-1}{2e}, \frac{e^{\frac{k-2}{k-1}}-1}{2e^{\frac{k-2}{k-1}}-1}\right\}$, *which is lower bounded by a constant $\frac{e-1}{2e}$.*

**Remark.** *The two terms of the approximation ratio in Theorem 1 have different advantages. While the first term remains constant (i.e., $\frac{e-1}{2e} \approx 0.316$), the second term (i.e., $(e^{\frac{k-2}{k-1}}-1)/(2e^{\frac{k-2}{k-1}}-1)$) is a monotonically increasing function of $k$. The first term is larger for a small value of $k$ (when $k < 4$); the second term becomes larger for a wide range of $k$ (when $k \geq 4$).*

In Theorem 2, we state the approximation result of Algorithm 1 in the case of $1 \leq \tau \leq k$.

**Theorem 2.** *Consider $1 \leq \tau \leq k$. Under Assumption 1, Algorithm 1 achieves an approximation ratio of* $\max\left\{\frac{(e-1)^2}{e(2e-1)}, \frac{(e-1)(e^{\frac{k-2\tau}{k-\tau}}-1)}{(2e-1)e^{\frac{k-2\tau}{k-\tau}}-(e-1)}\right\}$, *which is lower bounded by a constant $\frac{(e-1)^2}{e(2e-1)}$.*

**Remark.** *The two terms of the approximation ratio in Theorem 2 have different advantages. While the first term remains constant (i.e., $\frac{(e-1)^2}{e(2e-1)} \approx 0.245$), the second term is a monotonically increasing (resp., decreasing) function of $k$ (resp., $\tau$). The first term is larger when $k < \frac{2-\ln(\frac{e^2+e-1}{2e-1})}{1-\ln(\frac{e^2+e-1}{2e-1})}\tau$; the second term becomes larger when $k \geq \frac{2-\ln(\frac{e^2+e-1}{2e-1})}{1-\ln(\frac{e^2+e-1}{2e-1})}\tau$. We provide the approximation ratio under different values of $\tau$ and $k$ in Table **??** in the Appendix of the supplementary document.*

### 3.2 Robustness Against the Removal of Arbitrary Elements

In this subsection, we wish to design an algorithm that is robust against the removal of an arbitrary subset of $\tau$ selected elements, which are not necessarily contiguous. One weakness of Algorithm 1 is that the value of the selected sequence could be concentrated in the first few elements of subsequences $S_1$ and $S_2$. If we allow the removal of an arbitrary subset of $\tau$ selected elements, the removal of the first few elements of subsequences $S_1$ and $S_2$ could leave the remaining sequence with little or no value. By considering a special case in Section 3.1 where the removed elements are restricted to be contiguous, we have managed to prevent such worst case from happening. However, the problem

becomes more challenging when we consider a more general case without such a restriction. In the following, we propose an algorithm that is robust against the removal of an arbitrary subset of $\tau$ selected elements, but it requires a stronger assumption and achieves an approximation ratio that depends on the value of $\tau$. This new algorithm is presented in Algorithm 2.

Similar to Algorithm 1, Algorithm 2 works in two steps. However, there is a subtle yet critical difference in Step 1, which is the key to ensuring robustness in the general case. Specifically, in Step 1 of Algorithm 2, we select a sequence $S_1$ of $\tau$ elements from $\mathcal{V}$ by iteratively choosing an element $v$ in a greedy manner, based on its absolute value $h((v))$ instead of its marginal value $h((v)|S_1)$ as in Algorithm 1. We then select a sequence $S_2$ in Step 2, which is the same as that of Algorithm 1. The final output is $S = S_1 \oplus S_2$. Algorithm 2 also has a complexity of $O(kV)$.

Before we state the approximation results of Algorithm 2, we introduce a generalized definition of sequence submodularity. Function $h$ is said to be general-sequence-submodular if for all $S_3 \in \mathcal{H}(\mathcal{V})$, we have

$$h(S_3|S_1) \geq h(S_3|S_2), \ \forall S_1, S_2 \in \mathcal{H}(\mathcal{V}) \text{ such that } S_1 \text{ is a subsequence of } S_2. \tag{6}$$

Note that $S_1 \preceq S_2$ implies that $S_1$ is a subsequence of $S_2$, but not vice versa. Therefore, the general sequence submodularity defined above generalizes the sequence submodularity defined in Eq. (2) as a special case with $S_1 \preceq S_2$. Next, we state Assumption 2 and Theorem 3.

**Assumption 2.** *Function $h$ is forward-monotone, backward-monotone, and general-sequence-submodular.*

**Theorem 3.** *Consider $1 \leq \tau \leq k$. Under Assumption 2, Algorithm 2 achieves an approximation ratio of $\frac{1-1/e}{1+\tau}$.*

**Remark.** *While we only need the simplest form of the diminishing returns definition (element-sequence-submodularity) to establish provable approximation guarantees for the non-robust sequence submodular maximization, for its robust counterpart, we require stronger assumptions (sequence-submodularity and general-sequence-submodularity vs. element-sequence-submodularity) to show provable performance guarantees. In addition, consider a set function $r$ and ground set $\mathcal{V}$. While monotonicity of set function $r$ implies monotonicity of the same function with respect to the marginal value of adding a set to another set (i.e., $r(\mathcal{V}_2|\mathcal{V}_1) \triangleq r(\mathcal{V}_1 \cup \mathcal{V}_2) - r(\mathcal{V}_1)$ for any $\mathcal{V}_1, \mathcal{V}_2 \subseteq \mathcal{V}$), a similar property does not hold for backward monotonicity of sequence functions. This subtle difference results in a more involved analysis of showing similar results for sequence functions.*

## 4  Robust Approximate Sequence Submodular Maximization

In this section, we introduce generalized versions of sequence submodularity and backward monotonicity, which are called approximate sequence submodularity and approximate backward monotonicity. Then, we show that Algorithms 1 and 2 can also approximately solve Problem $(R)$ under weaker assumptions based on such generalized properties.

We begin with some additional definitions. Consider $\mu_1 \in (0, 1]$. Function $h$ is said to be $\mu_1$-element-sequence-submodular if for all $v \in \mathcal{V}$, we have

$$h((v)|S_1) \geq \mu_1 h((v)|S_2), \forall S_1, S_2 \in \mathcal{H}(\mathcal{V}) \text{ such that } S_1 \preceq S_2. \tag{7}$$

Also, consider $\mu_2 \in (0, 1]$. Function $h$ is said to be $\mu_2$-sequence-submodular if for all $S_3 \in \mathcal{H}(\mathcal{V})$, we have

$$h(S_3|S_1) \geq \mu_2 h(S_3|S_2), \ \forall S_1, S_2 \in \mathcal{H}(\mathcal{V}) \text{ such that } S_1 \preceq S_2. \tag{8}$$

Note that $\mu_1$ could be greater than $\mu_2$ for some function $h$. We distinguish $\mu_1$ and $\mu_2$ as some of our results depend on $\mu_1$ only. Similarly, consider $\mu_3 \in (0, 1]$. Function $h$ is said to be $\mu_3$-general-sequence-submodular if for all $S_3 \in \mathcal{H}(\mathcal{V})$, we have

$$h(S_3|S_1) \geq \mu_3 h(S_3|S_2), \ \forall S_1, S_2 \in \mathcal{H}(\mathcal{V}) \text{ such that } S_1 \text{ is a subsequence of } S_2. \tag{9}$$

Consider $\alpha \in (0, 1]$. Function $h$ is said to be $\alpha$-backward-monotone if

$$h(S_1 \oplus S_2) \geq \alpha h(S_2), \ \forall S_1, S_2 \in \mathcal{H}(\mathcal{V}). \tag{10}$$

Next, we state several assumptions that will be needed for deriving the main results in this section.

**Assumption 3.** *Function $h$ is forward-monotone, backward-monotone, $\mu_1$-element-sequence-submodular, and $\mu_2$-sequence-submodular.*

**Assumption 4.** *Function $h$ is forward-monotone, $\alpha$-backward-monotone, $\mu_1$-element-sequence-submodular, and $\mu_2$-sequence-submodular.*

**Assumption 5.** *Function $h$ is forward-monotone, $\alpha$-backward-monotone, $\mu_1$-element-sequence-submodular, and $\mu_3$-general-sequence-submodular.*

We are now ready to state the generalized approximation results of Algorithm 1 under Assumptions 3 and 4, respectively.

**Theorem 4.** *Consider $\tau = 1$. Under Assumption 3, Algorithm 1 achieves an approximation ratio of $\frac{a(e^b-1)}{e^b-a}$, where $a = \frac{\mu_1\mu_2}{\mu_1+1}$ and $b = \mu_1 \cdot \frac{k-2}{k-1}$; under Assumption 4, Algorithm 1 achieves an approximation ratio of $\frac{\alpha^2\mu_1\mu_2(e^{\mu_1}-1)}{(\mu_1+\alpha)e^{\mu_1}}$.*

**Theorem 5.** *Consider $1 \leq \tau \leq k$. Under Assumption 3, Algorithm 1 achieves an approximation ratio of $\frac{a\mu_2(e^b-1)}{(a+1)e^b-a\mu_2}$, where $a = \mu_1 \cdot (1 - 1/e^{\mu_1})$ and $b = \mu_1 \cdot \frac{k-2\tau}{k-\tau}$; under Assumption 4, Algorithm 1 achieves an approximation ratio of $\frac{\alpha^2\mu_1\mu_2(e^{\mu_1}-1)^2}{\mu_1 e^{\mu_1}(e^{\mu_1}-1)+e^{2\mu_1}}$.*

Finally, we state the approximation result of Algorithms 2 under Assumption 5.

**Theorem 6.** *Consider $1 \leq \tau \leq k$. Under Assumption 5, Algorithm 2 achieves an approximation ratio of $\frac{\alpha^2\mu_1\mu_3(e^{\mu_1}-1)}{(\mu_1+\alpha\tau)e^{\mu_1}}$.*

# 5 Related Work

Since the seminal work in [1], submodular maximization has been extensively studied in the literature. Several efficient approximation algorithms have been developed for maximizing a submodular set function in various settings (e.g., [1–4]). The concept of sequence (or string) submodularity for sequence functions is a generalization of submodularity, which has been introduced recently in several studies (e.g., [5–11, 26]). In [7], it has been shown that a simple greedy algorithm can achieve an approximation ratio of $(1 - 1/e)$ for maximizing a forward-monotone, backward-monotone, and element-sequence-submodular function.

On the other hand, robust versions of submodular maximization has been considered in some recent studies (e.g., [13–16]), where the focus is on selecting a set of elements that is robust against the removal of a subset of them. In [13], the authors propose the first algorithm with a constant approximation ratio for the problem of robust submodular maximization with cardinality constraints, where the selected set is of size $k$ and the robustness is against the removal of any $\tau$ elements of the selected set. The constant approximation ratio derived in [13] is valid as long as the number of removed elements is small compared to the selected set (i.e., $\tau = o(\sqrt{k})$). An extension that guarantees the same constant approximation ratio but allows the removal of a larger number of elements (i.e., $\tau = o(k)$) is presented in [14]. Another algorithm that allows the removal of an arbitrary number of elements under a mild assumption is presented in [15]. The work in [16] relaxes the restriction on $\tau$, but the achieved approximation ratio depends on the value of $\tau$. The work in [17] considers the same problem under different types of constraints, such as matroid and knapsack constraints. The work in [18, 19] extends the work in [16] to a multi-stage setting, where the decision at one stage takes into account the failures that happened in the previous stages. Other extensions that consider fairness and privacy issues are studied in [20, 21]. It is unclear whether all of these algorithms for robust set submodularity can be properly extended to our problem, as converting a set into a sequence could result in an arbitrarily bad performance. Even if so, it is more likely that establishing their approximation guarantees would require a more sophisticated analysis, which calls for more in-depth investigations. Note that the analysis of our simple greedy algorithms is already very sophisticated.

In [22], a different notion of robustness is considered, which is referred to as maximizing the minimum of multiple submodular functions. This work proposes a bicriterion approximation algorithm for the studied problem with cardinality constraints. Moreover, the work in [23–25] extends that of [22] to accommodate a wide variety of constraints, including matroid and knapsack constraints. The work in [27] develops an approximation algorithm for robust non-submodular maximization,

using other characterizations such as the submodularity ratio and the inverse curvature. The work in [12] introduces the idea of adaptive sequence submodular maximization, which aims to utilize the feedback obtained in previous iterations to improve the current decision. Note that while the work in [6, 12, 26] assumes that the sequential relationship among elements is encoded as a directed acyclic graph, we consider a general setting without such structures. It would indeed be interesting to explore our algorithms when the sequential relationship is encoded in a specific graphical form.

## 6 Conclusion

In this paper, we investigated a new problem of robust sequence submodular maximization. We discussed the unique challenges introduced by considering sequence functions and ensuring robustness guarantees. To address these novel challenges, we proposed two robust greedy algorithms and proved that they can achieve certain approximation ratios for the considered problem, assuming forward-monotone, backward-monotone, and sequence-submodular functions. We further introduced approximate versions of sequence submodularity and backward monotonicity and showed that the proposed algorithms can also provide performance guarantees under a larger class of weaker assumptions based on such generalized properties. Our future work includes developing more efficient algorithms with better approximation ratios in the general settings and investigating the possibility of obtaining similar results under the assumption of generalized/approximate forward monotonicity.

## 7 Broader Impact

This work contributes to the state-of-the-art theory of submodular optimization. The proposed algorithms and the presented approximation results can be applied to real-world applications where the stated assumptions of sequence submodularity and monotonicity or their approximate versions are satisfied. Several real-world applications, including machine learning based recommendation systems, ads allocation, and automation and control, involve the selection of elements in sequence.

## Acknowledgement

This work was supported in part by the NSF under Grants CNS-1651947, CNS-1824440, CNS-1828363, and CNS-1757533.

## Footnotes

[1]The approximation ratio is defined as the ratio of the objective value achieved by an algorithm over that achieved by an optimal algorithm.

[2]This definition can be easily generalized to allow repetition by augmenting the ground set $\mathcal{V}$ as follows. Assume that each element $v_i \in \mathcal{V}$ can be repeated $z_i$ times. Let $v_i^j$ denote the $j$-th copy of element $v_i$. We use $\bar{\mathcal{V}}$ to denote the augmented ground set, which is defined as $\bar{\mathcal{V}} \triangleq \cup_{v_i \in \mathcal{V}} \{v_i^1, \ldots, v_i^{z_i}\}$. Therefore, we can replace $\mathcal{V}$ with the augmented ground set $\bar{\mathcal{V}}$, which essentially allows the repetition of elements in $\mathcal{V}$.

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
