[Supplementary Material]

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

)\lvert S_1)$ <br> 5: **end while** <br> //**Step 2:** <br> 6: **while** $\lvert S_2 \rvert < k - \tau$ **do** <br> 7:    $S_2 =$ <br>    $S_2 \oplus \arg\max_{v \in \mathcal{V} \setminus (\mathcal{V}(S_1) \cup \mathcal{V}(S_2))} h((v)\lvert S_2)$ <br> 8: **end while** <br> 9: $S = S_1 \oplus S_2$ | 1: Input: $\mathcal{V}, k, \tau$; Output: $S$ <br> 2: Initialization: $S = S_1 = S_2 = ()$ <br> //**Step 1:** <br> 3: **while** $\lvert S_1 \rvert < \tau$ **do** <br> 4:    $S_1 = S_1 \oplus \arg\max_{v \in \mathcal{V} \setminus \mathcal{V}(S_1)} h((v))$ <br> 5: **end while** <br> //**Step 2:** <br> 6: **while** $\lvert S_2 \rvert < k - \tau$ **do** <br> 7:    $S_2 =$ <br>    $

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

# 8 Appendix

## 8.1 Element-sequence-submodular vs. Sequence-submodular: A Counterexample

In [5], it is noted (without a proof) that an element-sequence-submodular function must also be sequence-submodular. However, we find this claim false and present a counterexample in Table 2 to show that an element-sequence-submodular function is not necessarily sequence-submodular. In the presented example, it is easy to verify that function $h$ is element-sequence-submodular, i.e., Eq. (3) is satisfied; however, it is not sequence-submodular, i.e., Eq. (2) is not satisfied. Specifically, we have $h((v_2, v_3)|()) < h((v_2, v_3)|(v_1))$ due to $h((v_2, v_3)|()) = h((v_2, v_3)) - h(()) = 1.2 - 0 = 1.2$ and $h((v_2, v_3)|(v_1)) = h((v_1, v_2, v_3)) - h((v_1)) = 2.2 - 0.2 = 2$.

Table 2: A counterexample showing that an element-sequence-submodular function is not necessarily sequence-submodular, where $\mathcal{V} = \{v_1, v_2, v_3\}$ and the columns are input sequences, values of function $h$, and marginal values of $v_1$, $v_2$, and $v_3$ with respect to sequence $S$, respectively

| Sequence $S$ \ Function $h$ | $h(S)$ | $h((v_1)|S)$ | $h((v_2)|S)$ | $h((v_3)|S)$ |
|---|---|---|---|---|
| $()$ | 0 | 0.2 | 1.2 | 1 |
| $(v_1)$ | 0.2 | 0 | 1 | 1 |
| $(v_2)$ | 1.2 | 0 | 0 | 0 |
| $(v_3)$ | 1 | 0.2 | 1 | 0 |
| $(v_1, v_2)$ | 1.2 | 0 | 0 | 1 |
| $(v_2, v_1)$ | 1.2 | 0 | 0 | 0 |
| $(v_1, v_3)$ | 1.2 | 0 | 1 | 0 |
| $(v_3, v_1)$ | 1.2 | 0 | 1 | 0 |
| $(v_2, v_3)$ | 1.2 | 0 | 0 | 0 |
| $(v_3, v_2)$ | 2 | 0.2 | 0 | 0 |
| $(v_1, v_2, v_3)$ | 2.2 | 0 | 0 | 0 |
| $(v_1, v_3, v_2)$ | 2.2 | 0 | 0 | 0 |
| $(v_2, v_1, v_3)$ | 1.2 | 0 | 0 | 0 |
| $(v_2, v_3, v_1)$ | 1.2 | 0 | 0 | 0 |
| $(v_3, v_1, v_2)$ | 2.2 | 0 | 0 | 0 |
| $(v_3, v_2, v_1)$ | 2.2 | 0 | 0 | 0 |

## 8.2 Approximation Ratio of Theorem 2 with Different $\tau$ and $k$

Table 3: Approximation ratio of Theorem 2 with different values of $\tau$ and $k$

| $\tau$ \ $k$ | 50 | 52 | 54 | 56 | 58 | 60 | 62 | 64 | 66 | 68 |
|---|---|---|---|---|---|---|---|---|---|---|
| 2 | 0.28 | 0.281 | 0.281 | 0.281 | 0.281 | 0.281 | 0.281 | 0.282 | 0.282 | 0.282 |
| 4 | 0.275 | 0.275 | 0.275 | 0.276 | 0.276 | 0.277 | 0.277 | 0.277 | 0.277 | 0.278 |
| 6 | 0.268 | 0.269 | 0.269 | 0.27 | 0.271 | 0.271 | 0.272 | 0.272 | 0.273 | 0.273 |
| 8 | 0.26 | 0.261 | 0.262 | 0.263 | 0.264 | 0.265 | 0.266 | 0.267 | 0.267 | 0.268 |
| 10 | 0.25 | 0.252 | 0.254 | 0.255 | 0.257 | 0.258 | 0.259 | 0.261 | 0.262 | 0.262 |
| 12 | 0.245 | 0.245 | 0.245 | 0.246 | 0.248 | 0.25 | 0.252 | 0.253 | 0.255 | 0.256 |
| 14 | 0.245 | 0.245 | 0.245 | 0.245 | 0.245 | 0.245 | 0.245 | 0.245 | 0.247 | 0.249 |
| 16 | 0.245 | 0.245 | 0.245 | 0.245 | 0.245 | 0.245 | 0.245 | 0.245 | 0.245 | 0.245 |
| 18 | 0.245 | 0.245 | 0.245 | 0.245 | 0.245 | 0.245 | 0.245 | 0.245 | 0.245 | 0.245 |
| 20 | 0.245 | 0.245 | 0.245 | 0.245 | 0.245 | 0.245 | 0.245 | 0.245 | 0.245 | 0.245 |

## 8.3 Preliminary Results: Sequence Submodular Maximization

In this section, we introduce some additional notations and present some important preliminary results that will be used in the proofs of Theorems 1, 2, and 3.

---

**Algorithm 3** Sequence Submodular Greedy (SSG)

---

1: Input: elements $\mathcal{V}$, $k$; Output: $S$
2: Initialization: $S = ()$
3: **while** $|S| < k$ **do**
4:     $S = S \oplus \arg\max_{v \in \mathcal{V} \setminus \mathcal{V}(S)} h((v)|S)$
5: **end while**

---

We begin with the additional notations. We use sequence $S^*(\mathcal{V}, k, \tau)$ to denote an optimal solution to Problem $(R)$. Note that Problem $(P)$ is a special case of Problem $(R)$ with $\tau = 0$. Therefore, sequence $S^*(\mathcal{V}, k, 0)$ denotes an optimal solution to Problem $(P)$. For any given sequence $S \in \mathcal{H}(\mathcal{V})$, we use $\mathcal{Z}_\tau(S)$ to denote an optimal solution to the following minimization problem:

$$\min_{\mathcal{V}' \subseteq \mathcal{V}(S), |\mathcal{V}'| \leq \tau} h(S - \mathcal{V}'). \tag{11}$$

Let $g_\tau(S)$ denote the value of function $h$ with input $S$ that has elements in $\mathcal{Z}_\tau(S)$ removed, i.e., $g_\tau(S) \triangleq h(S - \mathcal{Z}_\tau(S))$. Therefore, the optimal value of Problem $(R)$ is $g_\tau(S^*(\mathcal{V}, k, \tau)) = h(S^*(\mathcal{V}, k, \tau) - \mathcal{Z}_\tau(S^*(\mathcal{V}, k, \tau)))$. Moreover, it is easy to see that we have

$$g_\tau(S^*(\mathcal{V}, k, \tau)) \leq h(S^*(\mathcal{V}, k - \tau, 0)). \tag{12}$$

This is because the left-hand side is the value of a feasible solution to Problem $(P)$ while the right-hand side is the value of an optimal solution to Problem $(P)$.

First, we restate the approximation performance of the SSG algorithm (presented in Algorithm 3) in [7]. Let sequence $S$ with $|S| = k$ be the one selected by the SSG algorithm, and let $S^i$ be the sequence consisting of the first $i$ elements of sequence $S$ with $1 \leq i \leq k$. The result in [7] is stated for the case of $i = k$ only, but it can be easily generalized for any $i$ with $1 \leq i \leq k$.

**Lemma 1** (Theorem 3 of [7]). *Consider $1 \leq i \leq k$. Under Assumption 1, we have $h(S^i) \geq (1 - 1/e^{\frac{i}{k}})h(S^*(\mathcal{V}, k, 0))$.*

Next, we state in Lemma 2 that the approximation performance of the SSG algorithm can be better characterized if the value of the selected sequence is concentrated in its first few elements. We present the proof of Lemma 2 in Appendix 8.4.

**Lemma 2.** *Consider $c \in (0, 1]$ and $1 \leq k' \leq k$. Suppose that the sequence selected by the SSG algorithm is $S$ with $|S| = k$ and that there exists a sequence $S_1$ with $|S_1| = k - k'$ such that $S_1 \preceq S$ and $h(S_1) \geq c \cdot h(S)$. Then, under Assumption 1, we have $h(S) \geq \frac{e^{\frac{k'}{k}} - 1}{e^{\frac{k'}{k}} - c} h(S^*(\mathcal{V}, k, 0))$.*

**Remark.** *Lemma 2 implies that additional information about the sequence selected by the SSG algorithm can be exploited to prove a better approximation ratio. In the following, we provide a few examples to demonstrate the significance of Lemma 2. Assume $k = 20$ and $k' = 19$. Then, we have $|S_1| = k - k' = 1$. If the value of sequence $S$ is evenly distributed over all of its twenty elements, then we have $c = 0.05$ and $h(S) \geq 0.62h(S^*(\mathcal{V}, k, 0))$. This approximation ratio is approximately equal to that in Lemma 1 (0.62 vs. $1 - 1/e \approx 0.63$). On the other hand, if the first element is worth a higher portion of the value of sequence $S$, then Lemma 2 leads to a better approximation ratio. For example, if $c = 0.5$ (resp., $c = 0.8$), then we have $h(S) \geq 0.76h(S^*(\mathcal{V}, k, 0))$ (resp., $h(S) \geq 0.88h(S^*(\mathcal{V}, k, 0))$). In particular, if $c = 1$, then we have $h(S) \geq h(S^*(\mathcal{V}, k, 0))$, which implies that the SSG algorithm yields an optimal solution to Problem $(P)$. We state this special case in Corollary 1, which may be of independent interest.*

**Corollary 1.** *Suppose that the sequence selected by the SSG algorithm is $S$ with $|S| = k$ and that there exists a sequence $S_1$ such that $S_1 \preceq S$ and $h(S_1) = h(S)$. Then, under Assumption 1, we have $h(S) = h(S^*(\mathcal{V}, k, 0))$, and thus, sequence $S$ is an optimal solution to Problem $(P)$.*

**Remark.** *Lemmas 1 and 2 and Corollary 1 can be proven in a similar manner under a weaker assumption that function $h$ is forward-monotone, backward-monotone, and element-sequence-submodular.*

In the following, we compare the impact of removing a certain number of elements from a selected sequence with that of removing the same number of elements from the ground set $\mathcal{V}$ before the selection takes place. We state the result in Lemma 3.

**Lemma 3.** *Consider $1 \leq \tau \leq k$. Suppose that function $h$ is forward-monotone. The following holds for any $\mathcal{V}' \subseteq \mathcal{V}$ with $|\mathcal{V}'| \leq \tau$:*

$$g_\tau(S^*(\mathcal{V}, k, \tau)) \leq h(S^*(\mathcal{V} \setminus \mathcal{V}', k - \tau, 0)). \tag{13}$$

*Proof.* Since function $h$ is forward-monotone, we can assume $|\mathcal{Z}_\tau(S)| = \tau$ for any sequence $S$ and $\tau \leq |S|$. The reason is the following. Suppose $|\mathcal{Z}_\tau(S)| < \tau$. Then, we can continue to remove more elements from the end of sequence $S$ till $|\mathcal{Z}_\tau(S)| = \tau$, which does not increase the value of the remaining sequence due to the forward monotonicity of function $h$.

Let $\mathcal{U} \triangleq \mathcal{V}(S^*(\mathcal{V}, k, \tau)) \cap \mathcal{V}'$ and $\tau' = |\mathcal{U}|$. Then, we have $\mathcal{U} \subseteq \mathcal{V}(S^*(\mathcal{V}, k, \tau))$. A little thoughts give $\mathcal{U} \cup \mathcal{Z}_{\tau - \tau'}(S^*(\mathcal{V}, k, \tau) - \mathcal{U}) \subseteq \mathcal{V}(S^*(\mathcal{V}, k, \tau))$ and $|\mathcal{U} \cup \mathcal{Z}_{\tau - \tau'}(S^*(\mathcal{V}, k, \tau) - \mathcal{U})| = \tau$. This implies that set $\mathcal{U} \cup \mathcal{Z}_{\tau - \tau'}(S^*(\mathcal{V}, k, \tau) - \mathcal{U})$ is a feasible solution to Problem (11) with respect to sequence $S^*(\mathcal{V}, k, \tau)$, which further implies the following:

$$h(S^*(\mathcal{V}, k, \tau) - \mathcal{Z}_\tau(S^*(\mathcal{V}, k, \tau))) \leq h(S^*(\mathcal{V}, k, \tau) - \mathcal{U} \cup \mathcal{Z}_{\tau - \tau'}(S^*(\mathcal{V}, k, \tau) - \mathcal{U})). \tag{14}$$

Also, from the definition of $\mathcal{U}$, we have $S^*(\mathcal{V}, k, \tau) - \mathcal{U} = S^*(\mathcal{V}, k, \tau) - \mathcal{V}'$, and thus, $\mathcal{Z}_{\tau - \tau'}(S^*(\mathcal{V}, k, \tau) - \mathcal{U}) = \mathcal{Z}_{\tau - \tau'}(S^*(\mathcal{V}, k, \tau) - \mathcal{V}')$. Again, from the definition of $\mathcal{U}$, we have that the elements in $\mathcal{V}' \setminus \mathcal{U}$ are not in sequence $S^*(\mathcal{V}, k, \tau)$. Then, we have the following:

$$h(S^*(\mathcal{V}, k, \tau) - \mathcal{U} \cup \mathcal{Z}_{\tau - \tau'}(S^*(\mathcal{V}, k, \tau) - \mathcal{U})) = h(S^*(\mathcal{V}, k, \tau) - \mathcal{V}' \cup \mathcal{Z}_{\tau - \tau'}(S^*(\mathcal{V}, k, \tau) - \mathcal{V}')). \tag{15}$$

Note that sequence $S^*(\mathcal{V}, k, \tau) - \mathcal{V}'$ does not contain any elements in $\mathcal{V}'$ and has $k - \tau'$ elements. Hence, sequence $S^*(\mathcal{V}, k, \tau) - \mathcal{V}'$ is a feasible solution to Problem $(R)$ (with respect to $\mathcal{V} \setminus \mathcal{V}'$, $k - \tau'$, and $\tau - \tau'$). This implies the following:

$$g_{\tau - \tau'}(S^*(\mathcal{V}, k, \tau) - \mathcal{V}') \leq g_{\tau - \tau'}(S^*(\mathcal{V} \setminus \mathcal{V}', k - \tau', \tau - \tau')). \tag{16}$$

Also, by replacing $\mathcal{V}$, $k$, and $\tau$ in Eq. (12) with $\mathcal{V} \setminus \mathcal{V}'$, $k - \tau'$, and $\tau - \tau'$, respectively, we immediately obtain the following:

$$g_{\tau - \tau'}(S^*(\mathcal{V} \setminus \mathcal{V}', k - \tau', \tau - \tau')) \leq h(S^*(\mathcal{V} \setminus \mathcal{V}', k - \tau, 0)). \tag{17}$$

By combining Eqs. (14)-(17), we have the following:

$$\begin{aligned}
g_\tau(S^*(\mathcal{V}, k, \tau)) &\overset{(a)}{=} h(S^*(\mathcal{V}, k, \tau) - \mathcal{Z}_\tau(S^*(\mathcal{V}, k, \tau))) \\
&\overset{(b)}{\leq} h(S^*(\mathcal{V}, k, \tau) - \mathcal{U} \cup \mathcal{Z}_{\tau - \tau'}(S^*(\mathcal{V}, k, \tau) - \mathcal{U})) \\
&\overset{(c)}{=} h(S^*(\mathcal{V}, k, \tau) - \mathcal{V}' \cup \mathcal{Z}_{\tau - \tau'}(S^*(\mathcal{V}, k, \tau) - \mathcal{V}')) \\
&\overset{(d)}{=} h((S^*(\mathcal{V}, k, \tau) - \mathcal{V}') - \mathcal{Z}_{\tau - \tau'}(S^*(\mathcal{V}, k, \tau) - \mathcal{V}')) \\
&\overset{(e)}{=} g_{\tau - \tau'}((S^*(\mathcal{V}, k, \tau) - \mathcal{V}')) \\
&\overset{(f)}{\leq} g_{\tau - \tau'}(S^*(\mathcal{V} \setminus \mathcal{V}', k - \tau', \tau - \tau')) \\
&\overset{(g)}{\leq} h(S^*(\mathcal{V} \setminus \mathcal{V}', k - \tau, 0)),
\end{aligned} \tag{18}$$

where (a) is from the definition of function $g_\tau$, (b) is from Eq. (14), (c) is from Eq. (15), (d) is a rewriting, (e) is from the definition of function $g_{\tau - \tau'}$, (f) is from Eq. (16), and (g) is from Eq. (17). $\quad\square$

## 8.4 Proof of Lemma 2

Before we prove Lemma 2, we introduce two lemmas: Lemmas 4 and 5. Lemma 4 is borrowed from [9] and will be used in the proof of Lemma 5, which will be used in the proof of Lemma 2.

**Lemma 4** (Lemma 10 of [9])**.** *Suppose that function $h$ is sequence-submodular. For any sequences $S_1', S_2' \in \mathcal{H}$, there exists an element $v \in \mathcal{V}$ such that $h((v)|S_1') \geq \frac{1}{|S_2'|} h(S_2'|S_1')$.*

**Lemma 5.** *Consider $1 \leq k' \leq k$. Suppose that the sequence selected by the SSG algorithm is $S$ with $|S| = k$ and that there exist sequences $S_1$ and $S_2$ such that sequence $S$ can be written as $S = S_1 \oplus S_2$ with $|S_1| = k - k'$ and $|S_2| = k'$. Then, under Assumption 1, we have $h(S_2|S_1) \geq (1 - 1/e^{\frac{k'}{k}})(h(S^*(\mathcal{V}, k, 0)) - h(S_1))$.*

*Proof.* Let $v_2^i$ denote the $i$-th element of sequence $S_2$, and let $S_2^i \triangleq (v_2^1, \ldots, v_2^i)$ denote the sequence consisting of the first $i$ elements of sequence $S_2$. Since function $h$ is forward-monotone, we can assume that $|S^*(\mathcal{V}, k, 0)| = k$ as adding more elements to the end of a sequence does not reduce its overall value.

Due to Lemma 4, there exists some element $v' \in \mathcal{V}$ such that $h((v')|S_1 \oplus S_2^{i-1}) \geq \frac{1}{k}h(S^*(\mathcal{V}, k, 0)|S_1 \oplus S_2^{i-1})$. Then, we have the following:

$$
\begin{aligned}
h(S_1 \oplus S_2^i) - h(S_1 \oplus S_2^{i-1}) &= h((v_2^i)|S_1 \oplus S_2^{i-1}) \\
&\overset{(a)}{\geq} h((v')|S_1 \oplus S_2^{i-1}) \\
&\overset{(b)}{\geq} \frac{1}{k}h(S^*(\mathcal{V}, k, 0)|S_1 \oplus S_2^{i-1}) \\
&= \frac{1}{k}(h(S_1 \oplus S_2^{i-1} \oplus S^*(\mathcal{V}, k, 0)) - h(S_1 \oplus S_2^{i-1})) \\
&\overset{(c)}{\geq} \frac{1}{k}(h(S^*(\mathcal{V}, k, 0)) - h(S_1 \oplus S_2^{i-1})),
\end{aligned}
\tag{19}
$$

where (a) is due to the greedy manner of the SSG algorithm (Line 4 of Algorithm 3), (b) is from the property of element $v'$ (due to Lemma 4), and (c) is due to the backward monotonicity of function $h$. Rewriting Eq. (19) yields the following equivalent inequality:

$$
h(S_1 \oplus S_2^i) \geq \frac{1}{k}h(S^*(\mathcal{V}, k, 0)) + (1 - \frac{1}{k})h(S_1 \oplus S_2^{i-1}). \tag{20}
$$

By writing Eq. (20) for $i \in \{1, \ldots, k'\}$ and combining them, we obtain the following:

$$
\begin{aligned}
h(S_1 \oplus S_2^{k'}) &\geq \sum_{j=0}^{k'-1} \frac{1}{k}(1 - \frac{1}{k})^j \, h(S^*(\mathcal{V}, k, 0)) + (1 - \frac{1}{k})^{k'} h(S_1) \\
&= (1 - (1 - \frac{1}{k})^{k'})h(S^*(\mathcal{V}, k, 0)) + (1 - \frac{1}{k})^{k'} h(S_1).
\end{aligned}
\tag{21}
$$

Applying Eq. (21) and the fact that $S_2 = S_2^{k'}$ yields the following:

$$
\begin{aligned}
h(S_2|S_1) &= h(S_1 \oplus S_2^{k'}) - h(S_1) \\
&\geq (1 - (1 - \frac{1}{k})^{k'})h(S^*(\mathcal{V}, k, 0)) + (1 - \frac{1}{k})^{k'} h(S_1) - h(S_1) \\
&= (1 - (1 - \frac{1}{k})^{k'})h(S^*(\mathcal{V}, k, 0)) - (1 - (1 - \frac{1}{k})^{k'})h(S_1) \\
&= (1 - (1 - \frac{1}{k})^{k'})(h(S^*(\mathcal{V}, k, 0)) - h(S_1)) \\
&\geq (1 - 1/e^{\frac{k'}{k}})(h(S^*(\mathcal{V}, k, 0)) - h(S_1)),
\end{aligned}
\tag{22}
$$

where the last inequality holds because $(1 - \frac{1}{k}) \leq e^{-\frac{1}{k}}$ and $h(S^*(\mathcal{V}, k, 0)) - h(S_1)$ is nonnegative. This completes the proof. $\qquad\square$

**Remark.** *We note a subtle yet critical difference between the monotonicity properties of set functions and sequence functions, which complicates the proof of Lemma 5. Consider a set function $r(\cdot)$. We define $r(\mathcal{V}_2|\mathcal{V}_1) \triangleq r(\mathcal{V}_1 \cup \mathcal{V}_2) - r(\mathcal{V}_1)$ as the marginal value of adding set $\mathcal{V}_2 \subseteq \mathcal{V}$ to another set $\mathcal{V}_1 \subseteq \mathcal{V}$. It is easy to see that the monotonicity of function $r(\cdot)$ implies the monotonicity of function $r(\cdot|\mathcal{V}_1)$ for any given $\mathcal{V}_1 \subseteq \mathcal{V}$. Such an analogy also exists for forward monotonicity of sequence functions, i.e., the forward monotonicity of a sequence function $h(\cdot)$ implies the forward monotonicity of function $h(\cdot|S_1)$ for any given $S_1 \in \mathcal{H}(\mathcal{V})$. However, a similar property does not hold for backward monotonicity. Due to lack of such a monotonicity property, the proof of Lemma 5 becomes more involved and requires more careful derivations.*

Having introduced Lemmas 4 and 5, we are now ready to prove Lemma 2.

*Proof of Lemma 2.* Suppose $h(S) = \delta \cdot h(S^*(\mathcal{V}, k, 0))$ for some $\delta \in (0, 1]$. Then, we have

$$
\begin{aligned}
\delta \cdot h(S^*(\mathcal{V}, k, 0)) &= h(S) \\
&= h(S_1) + h(S_2|S_1) \\
&\overset{(a)}{\geq} h(S_1) + (1 - 1/e^{\frac{k'}{k}})(h(S^*(\mathcal{V}, k, 0)) - h(S_1)) \\
&= (1/e^{\frac{k'}{k}})h(S_1) + (1 - 1/e^{\frac{k'}{k}})h(S^*(\mathcal{V}, k, 0)) \\
&\overset{(b)}{\geq} (c \cdot \delta/e^{\frac{k'}{k}})h(S^*(\mathcal{V}, k, 0)) + (1 - 1/e^{\frac{k'}{k}})h(S^*(\mathcal{V}, k, 0)),
\end{aligned}
\tag{23}
$$

where (a) follows from Lemma 5 and (b) holds because $h(S_1) \geq c \cdot h(S) = c \cdot \delta \cdot h(S^*(\mathcal{V}, k, 0))$. Dividing both sides of Eq. (23) by $h(S^*(\mathcal{V}, k, 0))$ yields the following:

$$
\delta \geq (c \cdot \delta/e^{\frac{k'}{k}}) + (1 - 1/e^{\frac{k'}{k}}),
\tag{24}
$$

which implies

$$
\delta \geq \frac{e^{\frac{k'}{k}} - 1}{e^{\frac{k'}{k}} - c}.
\tag{25}
$$

The above equation, along with $h(S) = \delta \cdot h(S^*(\mathcal{V}, k, 0))$, implies $h(S) \geq \frac{e^{\frac{k'}{k}} - 1}{e^{\frac{k'}{k}} - c} h(S^*(\mathcal{V}, k, 0))$. This completes the proof. $\qquad\square$

### 8.5 Proof of Theorem 1

*Proof.* Suppose that function $h$ is forward-monotone, backward-monotone, and sequence-submodular (Assumption 1). We use Lemmas 1, 2, and 3 presented in Appendix 8.3 to prove that Algorithm 1 achieves an approximation ratio of $\max \left\{ \frac{e-1}{2e}, \frac{e^{\frac{k-2}{k-1}} - 1}{2e^{\frac{k-2}{k-1}} - 1} \right\}$ in the case of $\tau = 1$.

Given $\tau = 1$, in Step 1 of Algorithm 1, the selected sequence $S_1$ consists of one element only; this element is denoted by $v_1$, i.e., $S_1 = (v_1)$. In Step 2 of Algorithm 1, it is equivalent that sequence $S_2$ is selected by the SSG algorithm from set $\mathcal{V} \setminus \{v_1\}$, and we have $|S_2| = k - \tau = k - 1$. Hence, the sequence selected by Algorithm 1 can be written as $S = S_1 \oplus S_2 = (v_1) \oplus S_2$. Recall that for any given sequence $S$, set $\mathcal{Z}_\tau(S)$ denotes the set of elements removed from sequence $S$ in the worst case (i.e., $\mathcal{Z}_\tau(S)$ is an optimal solution to Problem (11)). Note that only one element will be removed from $S$, i.e., $|\mathcal{Z}_\tau(S)| = 1$. For ease of notation, we use $z$ to denote the only element in $\mathcal{Z}_\tau(S)$, i.e., $\mathcal{Z}_\tau(S) = \{z\}$.

We want to show the following two bounds, which establish the approximation ratio of Algorithm 1:

$$
h(S - \{z\}) \geq \frac{e - 1}{2e} g_\tau(S^*(\mathcal{V}, k, \tau)),
\tag{26a}
$$

$$
h(S - \{z\}) \geq \frac{e^{\frac{k-2}{k-1}} - 1}{2e^{\frac{k-2}{k-1}} - 1} g_\tau(S^*(\mathcal{V}, k, \tau)).
\tag{26b}
$$

To begin with, we present a lower bound on $h(S_2)$, which will be used throughout the proof:

$$
\begin{aligned}
h(S_2) &\geq (1 - 1/e)h(S^*(\mathcal{V} \setminus \{v_1\}, k - \tau, 0)) \\
&\geq (1 - 1/e)g_\tau(S^*(\mathcal{V}, k, \tau)),
\end{aligned}
\tag{27}
$$

where the first inequality is from Lemma 1 (where we replace $\mathcal{V}$ with $\mathcal{V} \setminus \{v_1\}$ and both $k$ and $i$ with $k - \tau$) and the second inequality is from Lemma 3 (where we replace $\mathcal{V}'$ with $\{v_1\}$).

The proof proceeds as follows. Element $z$ is an element that will be removed, which can be either $v_1$ or an element in $S_2$. Therefore, we consider two cases: (I) $z = v_1$ and (II) $z \neq v_1$.

In Case I, we have $z = v_1$, which implies the following:

$$
h(S - \{z\}) = h(S_2) \geq (1 - 1/e)g_\tau(S^*(\mathcal{V}, k, \tau)),
\tag{28}
$$

where the inequality follows from Eq. (27).

In Case II, we have $z \neq v_1$ (or $z \in \mathcal{V}(S_2)$). Depending on the impact of removing element $z$, we consider two subcases: (II-a) $h(S_2) \leq h(S_2 - \{z\})$ and (II-b) $h(S_2) > h(S_2 - \{z\})$.

In Case II-a: we have $h(S_2) \leq h(S_2 - \{z\})$. In this case, the removal of element $z$ does not reduce the overall value of the remaining sequence $S_2 - \{z\}$. Then, we have

$$h(S - \{z\}) = h((v_1) \oplus (S_2 - \{z\})) \overset{(a)}{\geq} h(S_2 - \{z\}) \overset{(b)}{\geq} h(S_2) \overset{(c)}{\geq} (1 - 1/e)g_\tau(S^*(\mathcal{V}, k, \tau)), \quad (29)$$

where (a) is due to the backward monotonicity of function $h$, (b) holds from the condition of this subcase, and (c) follows from Eq. (27).

In Case II-b: we have $h(S_2) > h(S_2 - \{z\})$. Suppose $k = 2$. Then, it is trivial that the sequence selected by Algorithm 1 (i.e., $S = (v_1) \oplus (z)$) yields an optimal solution. This is because removing element $z$ from $S$ gives $(v_1)$, which has the largest individual value among all the elements. Therefore, we assume $k > 2$ throughout the rest of the proof. Let $\eta \triangleq \frac{h(S_2) - h(S_2 - \{z\})}{h(S_2)}$ denote the ratio of the loss of removing element $z$ from sequence $S_2$ to the value of sequence $S_2$, and we have $\eta \in (0, 1]$ due to $h(S_2) > h(S_2 - \{z\})$. We first state the following:

$$h(S - \{z\}) \geq \max\{\eta \cdot h(S_2), (1 - \eta) \cdot h(S_2)\}, \quad (30a)$$

$$\max\{\eta, (1 - \eta)\} \geq \frac{1}{2}, \quad (30b)$$

$$h(S_2) \geq \frac{e^{\frac{k-2}{k-1}} - 1}{e^{\frac{k-2}{k-1}} - \eta} g_\tau(S^*(\mathcal{V}, k, \tau)), \quad (30c)$$

$$\max\left\{\eta \cdot \frac{e^{\frac{k-2}{k-1}} - 1}{e^{\frac{k-2}{k-1}} - \eta}, (1 - \eta) \cdot \frac{e^{\frac{k-2}{k-1}} - 1}{e^{\frac{k-2}{k-1}} - \eta}\right\} \geq \frac{e^{\frac{k-2}{k-1}} - 1}{2(e^{\frac{k-2}{k-1}} - \frac{1}{2})}. \quad (30d)$$

We will prove Eqs. (30a)-(30d) later; for now, we assume that they all hold. Then, we can obtain the following bound:

$$
\begin{aligned}
h(S - \{z\}) &\geq \max\{\eta \cdot h(S_2), (1 - \eta) \cdot h(S_2)\} \\
&\geq \max\{\eta, 1 - \eta\} \cdot (1 - 1/e)g_\tau(S^*(\mathcal{V}, k, \tau)) \\
&\geq \frac{e - 1}{2e} g_\tau(S^*(\mathcal{V}, k, \tau)),
\end{aligned}
\quad (31)
$$

where the three inequalities are from Eqs. (30a), (27), and (30b), respectively.

Similarly, we can also obtain the following bound:

$$
\begin{aligned}
h(S - \{z\}) &\geq \max\{\eta \cdot h(S_2), (1 - \eta) \cdot h(S_2)\} \\
&\geq \max\left\{\frac{\eta(e^{\frac{k-2}{k-1}} - 1)}{e^{\frac{k-2}{k-1}} - \eta}, \frac{(1 - \eta)(e^{\frac{k-2}{k-1}} - 1)}{e^{\frac{k-2}{k-1}} - \eta}\right\} g_\tau(S^*(\mathcal{V}, k, \tau)) \\
&\geq \frac{e^{\frac{k-2}{k-1}} - 1}{2(e^{\frac{k-2}{k-1}} - \frac{1}{2})} g_\tau(S^*(\mathcal{V}, k, \tau)),
\end{aligned}
\quad (32)
$$

where the three inequalities are from Eqs. (30a), (30c), and (30d), respectively.

Combining all the cases establishes an approximation ratio of Algorithm 1 and completes the proof. Specifically, combining the bounds in Eqs. (28), (29), and (31) (resp., (32)) yields the bound in Eq. (26a) (resp., (26b)).

Now, it remains to show that Eqs. (30a)-(30d) hold in Case II-b, where we have $z \in \mathcal{V}(S_2)$ and $h(S_2) > h(S_2 - \{z\})$. We first rewrite $S_2$ as $S_2 = S_2^1 \oplus (z) \oplus S_2^2$, where $S_2^1$ and $S_2^2$ denote the subsequences of $S_2$ before and after element $z$, respectively. Note that $S_2^1$ or $S_2^2$ could be an empty

sequence, depending on the position of $z$ in $S_2$. Then, we characterize $h((z))$ in terms of $h(S_2)$:

$$
\begin{aligned}
\eta \cdot h(S_2) &\overset{(a)}{=} h(S_2) - h(S_2 - \{z\}) \\
&= h(S_2^1 \oplus (z) \oplus S_2^2) - h(S_2^1 \oplus S_2^2) \\
&= h(S_2^1) + h((z)|S_2^1) + h(S_2^2|S_2^1 \oplus (z)) - h(S_2^1) - h(S_2^2|S_2^1) \\
&= h((z)|S_2^1) + h(S_2^2|S_2^1 \oplus (z)) - h(S_2^2|S_2^1) \\
&\leq h((z)|S_2^1) \\
&\leq h((z)),
\end{aligned}
\tag{33}
$$

where (a) is from the definition of $\eta$ and the two inequalities are due to the sequence submodularity of function $h$. We are now ready to prove Eqs. (30a)-(30d).

To prove Eq. (30a), we decompose it into two parts: (i) $h(S - \{z\}) \geq \eta \cdot h(S_2)$ and (ii) $h(S - \{z\}) \geq (1 - \eta) \cdot h(S_2)$.

Part (i) can be shown through the following:

$$
h(S - \{z\}) \overset{(a)}{\geq} h((v_1)) \overset{(b)}{\geq} h((z)) \overset{(c)}{\geq} \eta \cdot h(S_2),
\tag{34}
$$

where (a) is form the forward monotonicity of function $h$, (b) is due to the greedy manner of Algorithm 1 (Lines 3-5), and (c) is from Eq. (33).

Part (ii) can be shown through the following:

$$
h(S - \{z\}) = h((v_1) \oplus (S_2 - \{z\})) \overset{(a)}{\geq} h(S_2 - \{z\}) \overset{(b)}{=} (1 - \eta) \cdot h(S_2),
$$

where (a) is from the backward monotonicity of function $h$ and (b) is from the definition of $\eta$.

Eq. (30b) holds trivially for any $\eta \in (0, 1]$ by setting $\eta$ and $1 - \eta$ to be equal and solving for $\eta$.

Next, we show that Eq. (30c) holds. Let $v_2^1$ denote the first element of sequence $S_2$. Then, we have the following:

$$
h((v_2^1)) \overset{(a)}{\geq} h((z)) \overset{(b)}{\geq} \eta \cdot h(S_2),
$$

where (a) holds because element $v_2^1$ has the largest individual value among all elements in $S_2$ and (b) follows from Eq. (33). Then, we can characterize the value of $h(S_2)$ as follows:

$$
\begin{aligned}
h(S_2) &\geq \frac{e^{\frac{k-2}{k-1}} - 1}{e^{\frac{k-2}{k-1}} - \eta} h(S^*(\mathcal{V} \setminus \{v_1\}, k - \tau, 0)) \\
&\geq \frac{e^{\frac{k-2}{k-1}} - 1}{e^{\frac{k-2}{k-1}} - \eta} g_\tau(S^*(\mathcal{V}, k, \tau)),
\end{aligned}
$$

where the first inequality is from Lemma 2 (where we replace $\mathcal{V}$, $S$, $S_1$, $k$, $k'$, and $c$ with $\mathcal{V} \setminus \{v_1\}$, $S_2$, $(v_2^1)$, $k - 1$, $k - 2$, and $\eta$, respectively) and the second inequality is from Lemma 3.

Finally, we show that Eq. (30d) holds. We define two functions of $\eta$: $l_1(\eta) \triangleq \eta \cdot \frac{e^{\frac{k-2}{k-1}} - 1}{e^{\frac{k-2}{k-1}} - \eta}$ and $l_2(\eta) \triangleq (1 - \eta) \cdot \frac{e^{\frac{k-2}{k-1}} - 1}{e^{\frac{k-2}{k-1}} - \eta}$. It is easy to verify that for $k > 2$ and $\eta \in (0, 1]$, function $l_1(\eta)$ is monotonically increasing and function $l_2(\eta)$ is monotonically decreasing. Also, we have $l_1(\frac{1}{2}) = l_2(\frac{1}{2}) = \frac{e^{\frac{k-2}{k-1}} - 1}{2(e^{\frac{k-2}{k-1}} - \frac{1}{2})}$. We consider two cases for $\eta$: $\eta \in [\frac{1}{2}, 1]$ and $\eta \in (0, \frac{1}{2}]$. For $\eta \in [\frac{1}{2}, 1]$, we have $\max\{l_1(\eta), l_2(\eta)\} \geq l_1(\eta) \geq l_1(\frac{1}{2})$ as $l_1(\eta)$ is monotonically increasing; for $\eta \in (0, \frac{1}{2}]$, we have $\max\{l_1(\eta), l_2(\eta)\} \geq l_2(\eta) \geq l_2(\frac{1}{2}) = l_1(\frac{1}{2})$ as $l_2(\eta)$ is monotonically decreasing. Therefore, for $\eta \in (0, 1]$, we have $\max\{l_1(\eta), l_2(\eta)\} \geq l_1(\frac{1}{2}) = \frac{e^{\frac{k-2}{k-1}} - 1}{2(e^{\frac{k-2}{k-1}} - \frac{1}{2})}$. This gives Eq. (30d) and completes the proof. $\qquad\square$

## 8.6 Proof of Theorem 2

*Proof.* Suppose that function $h$ is forward-monotone, backward-monotone, and sequence-submodular (Assumption 1). We use Lemmas 1, 2, and 3 presented in Appendix 8.3 to prove that Algorithm 1 achieves an approximation ratio of $\max\left\{\frac{(e-1)^2}{e(2e-1)}, \frac{(e-1)(e^{\frac{k-2\tau}{k-\tau}}-1)}{(2e-1)e^{\frac{k-2\tau}{k-\tau}}-(e-1)}\right\}$ in the case of $1 \leq \tau \leq k$, assuming the removal of $\tau$ contiguous elements.

In Step 1 of Algorithm 1, it is equivalent that sequence $S_1$ is selected by the SSG algorithm from set $\mathcal{V}$, and we have $|S_1| = \tau$. Similarly, in Step 2 of Algorithm 1, it is also equivalent that sequence $S_2$ is selected by the SSG algorithm from set $\mathcal{V} \setminus \mathcal{V}(S_1)$, and we have $|S_2| = k - \tau$. Hence, the sequence selected by Algorithm 1 can be written as $S = S_1 \oplus S_2$. Recall that for any given sequence $S$, set $\mathcal{Z}_\tau(S)$ denotes the set of elements removed from sequence $S$ in the worst case (i.e., $\mathcal{Z}_\tau(S)$ is an optimal solution to Problem (11)). We define $\mathcal{Z}_\tau^1(S) \triangleq \mathcal{Z}_\tau(S) \cap \mathcal{V}(S_1)$ and $\mathcal{Z}_\tau^2(S) \triangleq \mathcal{Z}_\tau(S) \cap \mathcal{V}(S_2)$ as the set of elements removed from subsequences $S_1$ and $S_2$, respectively.

The proof of Theorem 2 follows a similar line of analysis as in the proof of Theorem 1 for the case of $\tau = 1$. Specifically, we will also consider three cases: (I) $\mathcal{Z}_\tau^2(S) = \emptyset$, (II-a) $\mathcal{Z}_\tau^2(S) \neq \emptyset$ and $h(S_2) \leq h(S_2 - \mathcal{Z}_\tau^2(S))$, and (II-b) $\mathcal{Z}_\tau^2(S) \neq \emptyset$ and $h(S_2) > h(S_2 - \mathcal{Z}_\tau^2(S))$. The proofs of Case I and Case II-a are almost the same as those in Theorem 1, except for some minor technical differences. However, the proof of Case II-b is substantially different. The reason is the following. In the proof of Theorem 1, only one element can be removed from $S$, and in Case II-b, this element has to be in $\mathcal{V}(S_2)$, which makes it easier to characterize the impact of removing such an element. In the case of $1 \leq \tau \leq k$, however, more than one element may be removed, which could be in either $\mathcal{V}(S_1)$ or $\mathcal{V}(S_2)$ or both. Therefore, we present a different approach to address this new technical challenge.

We want to show the following two bounds that establish the approximation ratio of Algorithm 1:

$$h(S - \mathcal{Z}_\tau(S)) \geq \frac{(e-1)^2}{e(2e-1)} g_\tau(S^*(\mathcal{V}, k, \tau)), \tag{35a}$$

$$h(S - \mathcal{Z}_\tau(S)) \geq \frac{(e-1)(e^{\frac{k-2\tau}{k-\tau}}-1)}{(2e-1)e^{\frac{k-2\tau}{k-\tau}}-(e-1)} g_\tau(S^*(\mathcal{V}, k, \tau)). \tag{35b}$$

To begin with, we present a lower bound on $h(S_2)$, which will be used throughout the proof:

$$\begin{aligned} h(S_2) &\geq (1 - 1/e)h(S^*(\mathcal{V} \setminus \mathcal{V}(S_1), k - \tau, 0)) \\ &\geq (1 - 1/e)g_\tau(S^*(\mathcal{V}, k, \tau)), \end{aligned} \tag{36}$$

where the first inequality is from Lemma 1 (where we replace $\mathcal{V}$, $k$, and $i$ with $\mathcal{V} \setminus \mathcal{V}(S_1)$, $k - \tau$, and $k - \tau$, respectively) and the second inequality is from Lemma 3 (where we replace $\mathcal{V}'$ with $\mathcal{V}(S_1)$).

The proof proceeds as follows. Elements in $\mathcal{Z}_\tau(S)$ will be removed from sequence $S$. These elements can be either fully or partially in $\mathcal{V}(S_1)$ (i.e., $\mathcal{Z}_\tau^2(S) = \emptyset$ or $\mathcal{Z}_\tau^2(S) \neq \emptyset$). Therefore, we consider two cases: (I) $\mathcal{Z}_\tau^2(S) = \emptyset$ and (II) $\mathcal{Z}_\tau^2(S) \neq \emptyset$.

In Case I, we have $\mathcal{Z}_\tau^2(S) = \emptyset$, i.e., $\mathcal{Z}_\tau(S) = \mathcal{V}(S_1)$. Then, we have the following:

$$h(S - \mathcal{Z}_\tau(S)) = h(S_2) \geq (1 - 1/e)g_\tau(S^*(\mathcal{V}, k, \tau)), \tag{37}$$

where the inequality follows from Eq. (36).

In Case II, we have $\mathcal{Z}_\tau^2(S) \neq \emptyset$. Depending on the impact of removing elements in $\mathcal{Z}_\tau^2(S)$, we consider two subcases: (II-a) $h(S_2) \leq h(S_2 - \mathcal{Z}_\tau^2(S))$ and (II-b) $h(S_2) > h(S_2 - \mathcal{Z}_\tau^2(S))$.

In Case II-a: we have $h(S_2) \leq h(S_2 - \mathcal{Z}_\tau^2(S))$. In this case, the removal of elements in $\mathcal{Z}_\tau^2(S)$ does not reduce the overall value of the remaining sequence $S_2 - \mathcal{Z}_\tau^2(S)$. Then, we have

$$\begin{aligned} h(S - \mathcal{Z}_\tau(S)) &= h((S_1 - \mathcal{Z}_\tau^1(S)) \oplus (S_2 - \mathcal{Z}_\tau^2(S))) \\ &\overset{(a)}{\geq} h(S_2 - \mathcal{Z}_\tau^2(S)) \\ &\overset{(b)}{\geq} h(S_2) \\ &\overset{(c)}{\geq} (1 - 1/e)g_\tau(S^*(\mathcal{V}, k, \tau)), \end{aligned} \tag{38}$$

where (a) is due to the backward monotonicity of function $h$, (b) holds from the condition of this subcase, and (c) follows from Eq. (36).

In Case II-b: we have $h(S_2) > h(S_2 - \mathcal{Z}_\tau^2(S))$. Let $\tau_1 \triangleq |\mathcal{Z}_\tau^1(S)|$ and $\tau_2 \triangleq |\mathcal{Z}_\tau^2(S)|$. Then, we have $\tau = \tau_1 + \tau_2$ and $k = |S| = |S_1| + |S_2| \geq \tau + \tau_2$. We consider two cases: $k = \tau + \tau_2$ and $k > \tau + \tau_2$.

Suppose $k = \tau + \tau_2$. Then, it implies $\mathcal{Z}_\tau^2(S) = \mathcal{V}(S_2)$, i.e., all the elements in $S_2$ are removed. This further implies that the elements in $\mathcal{Z}_\tau^1(S)$ are at the end of sequence $S_1$ (due to the contiguous assumption of elements in $\mathcal{Z}_\tau(S)$). Let $S_1^{\tau_2} \preceq S_1$ denote the subsequence consisting of the first $\tau_2$ elements in $S_1$. It is easy to see $S_1 - \mathcal{Z}_\tau^1(S) = S_1^{\tau_2}$. Then, we have the following:

$$
\begin{aligned}
h(S - \mathcal{Z}_\tau(S)) &= h((S_1 - \mathcal{Z}_\tau^1(S)) \oplus (S_2 - \mathcal{Z}_\tau^2(S))) \\
&= h(S_1 - \mathcal{Z}_\tau^1(S)) \\
&= h(S_1^{\tau_2}) \\
&\geq (1 - 1/e)h(S^*(\mathcal{V}, \tau_2, 0)) \\
&\geq (1 - 1/e)g_\tau(S^*(\mathcal{V}, k, \tau)),
\end{aligned}
\tag{39}
$$

where the first inequality is from Lemma 1 (where we replace both $k$ and $i$ with $\tau_2$) and the second inequality is due to $\tau_2 = k - \tau$ and Lemma 3 (where $\mathcal{V}'$ is an empty set).

Now, suppose $k > \tau + \tau_2$. Let $\eta \triangleq \frac{h(S_2) - h(S_2 - \mathcal{Z}_\tau^2(S))}{h(S_2)}$ denote the ratio of the loss of removing elements in $\mathcal{Z}_\tau^2(S)$ from sequence $S_2$ to the value of sequence $S_2$, and we have $\eta \in (0, 1]$ due to $h(S_2) > h(S_2 - \mathcal{Z}_\tau^2(S))$. We first state the following:

$$
h(S - \mathcal{Z}_\tau(S)) \geq \max\{\eta(1 - 1/e)h(S_2), (1 - \eta)h(S_2)\},
\tag{40a}
$$

$$
\max\{\eta(1 - 1/e), (1 - \eta)\} \geq \frac{e - 1}{2e - 1},
\tag{40b}
$$

$$
h(S_2) \geq \frac{e^{\frac{k - \tau - \tau_2}{k - \tau_2}} - 1}{e^{\frac{k - \tau - \tau_2}{k - \tau_2}} - \eta(1 - 1/e)} g_\tau(S^*(\mathcal{V}, k, \tau)),
\tag{40c}
$$

$$
\max\left\{\frac{\eta(1 - 1/e)(e^{\frac{k - \tau - \tau_2}{k - \tau_2}} - 1)}{e^{\frac{k - \tau - \tau_2}{k - \tau_2}} - \eta(1 - 1/e)}, \frac{(1 - \eta)(e^{\frac{k - \tau - \tau_2}{k - \tau_2}} - 1)}{e^{\frac{k - \tau - \tau_2}{k - \tau_2}} - \eta(1 - 1/e)}\right\} \geq \frac{(e - 1)(e^{\frac{k - \tau - \tau_2}{k - \tau_2}} - 1)}{(2e - 1)e^{\frac{k - \tau - \tau_2}{k - \tau_2}} - (e - 1)}.
\tag{40d}
$$

We will prove Eqs. (40a)-(40d) later; for now, we assume that they all hold. Then, we can obtain the following bound:

$$
\begin{aligned}
h(S - \mathcal{Z}_\tau(S)) &\geq \max\{\eta(1 - 1/e)h(S_2), (1 - \eta)h(S_2)\} \\
&\geq \max\{\eta(1 - 1/e), (1 - \eta)\} \cdot (1 - 1/e)g_\tau(S^*(\mathcal{V}, k, \tau)) \\
&\geq \frac{(e - 1)^2}{e(2e - 1)} g_\tau(S^*(\mathcal{V}, k, \tau)),
\end{aligned}
\tag{41}
$$

where the three inequalities are from Eqs. (40a), (36), and (40b), respectively.

Similarly, we can also obtain the following bound:

$$
\begin{aligned}
h(S - \mathcal{Z}_\tau(S)) &\geq \max\{\eta(1 - 1/e)h(S_2), (1 - \eta)h(S_2)\} \\
&\geq \max\left\{\frac{\eta(e - 1)(e^{\frac{k - \tau - \tau_2}{k - \tau_2}} - 1)}{e(e^{\frac{k - \tau - \tau_2}{k - \tau_2}} - \eta(1 - 1/e))}, \frac{(1 - \eta)(e^{\frac{k - \tau - \tau_2}{k - \tau_2}} - 1)}{e^{\frac{k - \tau - \tau_2}{k - \tau_2}} - \eta(1 - 1/e)}\right\} g_\tau(S^*(\mathcal{V}, k, \tau)) \\
&\geq \frac{(e - 1)(e^{\frac{k - \tau - \tau_2}{k - \tau_2}} - 1)}{(2e - 1)e^{\frac{k - \tau - \tau_2}{k - \tau_2}} - (e - 1)} g_\tau(S^*(\mathcal{V}, k, \tau)) \\
&\geq \frac{(e - 1)(e^{\frac{k - 2\tau}{k - \tau}} - 1)}{(2e - 1)e^{\frac{k - 2\tau}{k - \tau}} - (e - 1)} g_\tau(S^*(\mathcal{V}, k, \tau)),
\end{aligned}
\tag{42}
$$

where the first three inequalities are from Eqs. (40a), (40c), and (40d), respectively, and the last inequality is due to $\tau_2 \leq \tau$ and that $\frac{(e-1)(e^{\frac{k-\tau-\tau_2}{k-\tau_2}}-1)}{(2e-1)e^{\frac{k-\tau-\tau_2}{k-\tau_2}}-(e-1)}$ is a decreasing function of $\tau_2$.

Combining all the cases establishes the approximation ratios of Algorithm 1 and completes the proof. Specifically, combining the bounds in Eqs. (37), (38), (39), and (41) (resp., (42)) yields the bound in Eq. (35a) (resp., (35b)).

Now, it remains to show that Eqs. (40a)-(40d) hold in Case II-b, where we have $\mathcal{Z}_\tau^2(S) \neq \emptyset$ and $h(S_2) > h(S_2 - \mathcal{Z}_\tau^2(S))$. Recall that the $\tau$ elements in $\mathcal{Z}_\tau(S)$ form a contiguous subsequence of $S$. Then, the elements in $\mathcal{Z}_\tau^1(S)$ and $\mathcal{Z}_\tau^2(S)$ also form a contiguous subsequence of $S_1$ and $S_2$, respectively. We use $Z_1$ and $Z_2$ to denote the contiguous subsequence of elements in $\mathcal{Z}_\tau^1(S)$ and $\mathcal{Z}_\tau^2(S)$, respectively. We first rewrite $S_2$ as $S_2 = S_2^1 \oplus Z_2 \oplus S_2^2$, where $S_2^1$ and $S_2^2$ denote the subsequences of $S_2$ before and after subsequence $Z_2$, respectively. Note that $S_2^1$ or $S_2^2$ could be an empty sequence, depending on the position of subsequence $Z_2$ in $S_2$. Then, we characterize $h(Z_2)$ in terms of $h(S_2)$:

$$
\begin{aligned}
\eta \cdot h(S_2) &\overset{(a)}{=} h(S_2) - h(S_2 - \mathcal{Z}_\tau^2(S)) \\
&= h(S_2^1 \oplus Z_2 \oplus S_2^2) - h(S_2^1 \oplus S_2^2) \\
&= h(S_2^1) + h(Z_2|S_2^1) + h(S_2^2|S_2^1 \oplus Z_2) - h(S_2^1) - h(S_2^2|S_2^1) \\
&= h(Z_2|S_2^1) + h(S_2^2|S_2^1 \oplus Z_2) - h(S_2^2|S_2^1) \\
&\leq h(Z_2|S_2^1) \\
&\leq h(Z_2),
\end{aligned}
\tag{43}
$$

where (a) is from the definition of $\eta$ and the two inequalities are due the sequence submodularity of function $h$. We are now ready to prove Eqs. (40a)-(40d).

To prove Eq. (40a), we decompose it into two parts: (i) $h(S - \mathcal{Z}_\tau(S)) \geq \eta(1 - 1/e)h(S_2)$ and (ii) $h(S - \mathcal{Z}_\tau(S)) \geq (1 - \eta)h(S_2)$.

Recall that $S_1^{\tau_2}$ denote the subsequence consisting of the first $\tau_2$ elements in $S_1$. Then, Part (i) can be shown through the following:

$$
\begin{aligned}
h(S - \mathcal{Z}_\tau(S)) &= h((S_1 - \mathcal{Z}_\tau^1(S)) \oplus (S_2 - \mathcal{Z}_\tau^2(S))) \\
&\overset{(a)}{\geq} h(S_1 - \mathcal{Z}_\tau^1(S)) \\
&= h(S_1^{\tau_2}) \\
&\overset{(b)}{\geq} (1 - 1/e)h(S^*(\mathcal{V}, \tau_2, 0)) \\
&\overset{(c)}{\geq} (1 - 1/e)h(Z_2) \\
&\overset{(d)}{\geq} \eta(1 - 1/e)h(S_2),
\end{aligned}
\tag{44}
$$

where (a) is due to the forward monotonicity of function $h$, (b) is due to the greedy manner of selecting subsequence $S_1^{\tau_2}$ from set $\mathcal{V}$ and Lemma 1 (where we replace both $k$ and $i$ with $\tau_2$), (c) holds because sequence $Z_2$ is a feasible solution to Problem $(P)$ for selecting a sequence of $\tau_2$ elements from $\mathcal{V}$, and (d) is from Eq. (43).

Part (ii) can be shown through the following:

$$
\begin{aligned}
h(S - \mathcal{Z}_\tau(S)) &= h((S_1 - \mathcal{Z}_\tau^1(S)) \oplus (S_2 - \mathcal{Z}_\tau^2(S))) \\
&\overset{(a)}{\geq} h(S_2 - \mathcal{Z}_\tau^2(S)) \\
&\overset{(b)}{=} (1 - \eta)h(S_2),
\end{aligned}
\tag{45}
$$

where (a) is from the backward monotonicity of function $h$ and (b) is from the definition of $\eta$.

Eq. (40b) holds trivially for any $\eta \in (0, 1]$ by setting $\eta(1 - 1/e)$ and $1 - \eta$ to be equal, solving for $\eta$, and plugging it back.

Next, we show that Eq. (40c) holds. Let $S_2^{\tau_2} \preceq S_2$ denote the subsequence consisting of the first $\tau_2$ elements of sequence $S_2$. Then, we have the following:

$$
\begin{aligned}
h(S_2^{\tau_2}) &\overset{(a)}{\geq} (1 - 1/e)h(S^*(\mathcal{V} \setminus \mathcal{V}(S_1), \tau_2, 0)) \\
&\overset{(b)}{\geq} (1 - 1/e)h(Z_2) \\
&\overset{(c)}{\geq} \eta(1 - 1/e)h(S_2),
\end{aligned}
\tag{46}
$$

where (a) is due to the greedy manner of selecting subsequence $S_2^{\tau_2}$ from set $\mathcal{V} \setminus \mathcal{V}(S_1)$ and Lemma 1 (where we replace $\mathcal{V}$, $k$, and $i$ with $\mathcal{V} \setminus \mathcal{V}(S_1)$, $\tau_2$, and $\tau_2$, respectively), (b) holds because sequence $Z_2$ is a feasible solution to Problem $(P)$ for selecting a sequence of $\tau_2$ elements from $\mathcal{V} = \mathcal{V} \setminus \mathcal{V}(S_1)$, and (c) is from Eq. (43). Therefore, we can characterize the value of $h(S_2)$ as follows:

$$
\begin{aligned}
h(S_2) &\geq \frac{e^{\frac{k-\tau-\tau_2}{k-\tau_2}} - 1}{e^{\frac{k-\tau-\tau_2}{k-\tau_2}} - \eta(1 - 1/e)} h(S^*(\mathcal{V} \setminus \mathcal{V}(S_1), k - \tau, 0)) \\
&\geq \frac{e^{\frac{k-\tau-\tau_2}{k-\tau_2}} - 1}{e^{\frac{k-\tau-\tau_2}{k-\tau_2}} - \eta(1 - 1/e)} g_\tau(S^*(\mathcal{V}, k, \tau)),
\end{aligned}
\tag{47}
$$

where the first inequality is from Lemma 2 (where we replace $\mathcal{V}$, $S$, $S_1$, $k$, $k'$, and $c$ with $\mathcal{V} \setminus \mathcal{V}(S_1)$, $S_2$, $S_2^{\tau_2}$, $k - \tau$, $k - \tau - \tau_2$, and $\eta(1 - 1/e)$, respectively), and the second inequality is from Lemma 3.

Finally, we show that Eq. (40d) holds. We define two functions of $\eta$:

$$
l_1(\eta) = \eta(1 - 1/e) \cdot \frac{e^{\frac{k-\tau-\tau_2}{k-\tau_2}} - 1}{e^{\frac{k-\tau-\tau_2}{k-\tau_2}} - \eta(1 - 1/e)} \quad \text{and} \quad l_2(\eta) = (1 - \eta) \cdot \frac{(e^{\frac{k-\tau-\tau_2}{k-\tau_2}} - 1)}{e^{\frac{k-\tau-\tau_2}{k-\tau_2}} - \eta(1 - 1/e)}.
$$

It is easy to verify that for $k > \tau + \tau_2$ and $\eta \in (0, 1]$, function $l_1(\eta)$ is monotonically increasing and function $l_2(\eta)$ is monotonically decreasing. Also, we have $l_1(\frac{e}{2e-1}) = l_2(\frac{e}{2e-1}) = \frac{(e-1)(e^{\frac{k-\tau-\tau_2}{k-\tau_2}} - 1)}{(2e-1)e^{\frac{k-\tau-\tau_2}{k-\tau_2}} - (e-1)}$. We consider two cases for $\eta$: $\eta \in [\frac{e}{2e-1}, 1]$ and $\eta \in (0, \frac{e}{2e-1}]$. For $\eta \in [\frac{e}{2e-1}, 1]$, we have $\max\{l_1(\eta), l_2(\eta)\} \geq l_1(\eta) \geq l_1(\frac{e}{2e-1})$ as $l_1(\eta)$ is monotonically increasing; for $\eta \in (0, \frac{e}{2e-1}]$, we have $\max\{l_1(\eta), l_2(\eta)\} \geq l_2(\eta) \geq l_2(\frac{e}{2e-1}) = l_1(\frac{e}{2e-1})$ as $l_2(\eta)$ is monotonically decreasing. Therefore, for $\eta \in (0, 1]$, we have $\max\{l_1(\eta), l_2(\eta)\} \geq l_1(\frac{e}{2e-1}) = \frac{(e-1)(e^{\frac{k-\tau-\tau_2}{k-\tau_2}} - 1)}{(2e-1)e^{\frac{k-\tau-\tau_2}{k-\tau_2}} - (e-1)}$. This gives Eq. (40d) and completes the proof. $\qquad\square$

## 8.7 Proof of Theorem 3

*Proof.* Suppose that function $h$ is forward-monotone, backward-monotone, and general-sequence-submodular (Assumption 2). We use Lemmas 1 and 3 presented in Appendix 8.3 to prove that Algorithm 2 achieves an approximation ratio of $\frac{1-1/e}{1+\tau}$ in the case of $1 \leq \tau \leq k$, assuming the removal of an arbitrary subset of $\tau$ selected elements, which are not necessarily contiguous.

In Step 1 of Algorithm 2, sequence $S_1$ is selected by choosing $\tau$ elements with the highest individual values in a greedy manner, and we have $|S_1| = \tau$. In Step 2 of Algorithm 1, it is equivalent that sequence $S_2$ is selected by the SSG algorithm from set $\mathcal{V} \setminus \mathcal{V}(S_1)$, and we have $|S_2| = k - \tau$. Hence, the sequence selected by Algorithm 1 can be written as $S = S_1 \oplus S_2$. Recall that for any given sequence $S$, set $\mathcal{Z}_\tau(S)$ denotes the set of elements removed from sequence $S$ in the worst case (i.e., $\mathcal{Z}_\tau(S)$ is an optimal solution to Problem (11)). We define $\mathcal{Z}_\tau^1(S) \triangleq \mathcal{Z}_\tau(S) \cap \mathcal{V}(S_1)$ and $\mathcal{Z}_\tau^2(S) \triangleq \mathcal{Z}_\tau(S) \cap \mathcal{V}(S_2)$ as the set of elements removed from subsequences $S_1$ and $S_2$, respectively.

The proof of Theorem 3 follows a similar line of analysis as in the proof of Theorem 2. Specifically, we will also consider three cases: (I) $\mathcal{Z}_\tau^2(S) = \emptyset$, (II-a) $\mathcal{Z}_\tau^2(S) \neq \emptyset$ and $h(S_2) \leq h(S_2 - \mathcal{Z}_\tau^2(S))$, and (II-b) $\mathcal{Z}_\tau^2(S) \neq \emptyset$ and $h(S_2) > h(S_2 - \mathcal{Z}_\tau^2(S))$. The proofs of Case I and Case II-a are identical to those in Theorem 2. Therefore, we focus on Case II-b, which requires a different proof strategy.

We want to show the following bound that establishes the approximation ratio of Algorithm 2:

$$h(S - \mathcal{Z}_\tau(S)) \geq \frac{1 - 1/e}{1 + \tau} g_\tau(S^*(\mathcal{V}, k, \tau)). \tag{48}$$

To begin with, we present a lower bound on $h(S_2)$, which will be used throughout the proof:

$$\begin{aligned}
h(S_2) &\geq (1 - 1/e)h(S^*(\mathcal{V} \setminus \mathcal{V}(S_1), k - \tau, 0)) \\
&\geq (1 - 1/e)g_\tau(S^*(\mathcal{V}, k, \tau)),
\end{aligned} \tag{49}$$

where the first inequality is from Lemma 1 (where we replace $\mathcal{V}$, $k$, and $i$ with $\mathcal{V} \setminus \mathcal{V}(S_1)$, $k - \tau$, and $k - \tau$, respectively) and the second inequality is from Lemma 3 (where we replace $\mathcal{V}'$ with $\mathcal{V}(S_1)$).

We now focus on Case II-b, where we have $\mathcal{Z}_\tau^2(S) \neq \emptyset$ and $h(S_2) > h(S_2 - \mathcal{Z}_\tau^2(S))$. Let $\eta \triangleq \frac{h(S_2) - h(S_2 - \mathcal{Z}_\tau^2(S))}{h(S_2)}$ denote the ratio of the loss of removing elements in $\mathcal{Z}_\tau^2(S)$ from sequence $S_2$ to the value of sequence $S_2$, and we have $\eta \in (0, 1]$ due to $h(S_2) > h(S_2 - \mathcal{Z}_\tau^2(S))$. We first state the following:

$$h(S - \mathcal{Z}_\tau(S)) \geq \max\{\frac{\eta}{\tau} \cdot h(S_2), (1 - \eta) \cdot h(S_2)\}, \tag{50a}$$

$$\max\{\frac{\eta}{\tau}, (1 - \eta)\} \geq \frac{1}{\tau + 1}. \tag{50b}$$

We will prove Eqs. (50a) and (50b) later; for now, we assume that they both hold. Then, we can obtain the following:

$$\begin{aligned}
h(S - \mathcal{Z}_\tau(S)) &\geq \max\{\frac{\eta}{\tau} \cdot h(S_2), (1 - \eta) \cdot h(S_2)\} \\
&\geq \max\{\frac{\eta}{\tau}, (1 - \eta)\} \cdot (1 - 1/e)g_\tau(S^*(\mathcal{V}, k, \tau)) \\
&\geq \frac{1 - 1/e}{\tau + 1} g_\tau(S^*(\mathcal{V}, k, \tau)),
\end{aligned} \tag{51}$$

where the three inequalities are from Eqs. (50a), (49), and (50b), respectively.

Now, we show that Eqs. (50a) and (50b) hold. We start by characterizing the value of elements in $\mathcal{Z}_\tau^2(S)$. Let $\tau_2 \triangleq |\mathcal{Z}_\tau^2(S)|$, and let the elements in $\mathcal{Z}_\tau^2(S)$ be denoted by $z_1, z_2, \ldots, z_{\tau_2}$ according to their order in sequence $S_2$. Then, we can rewrite $S_2$ as $S_2 = S_2^1 \oplus (z_1) \oplus S_2^2 \oplus (z_2) \oplus \cdots \oplus S_2^{\tau_2 + 1}$, where $S_2^i$ is the subsequence between elements $z_{i-1}$ and $z_i$, for $i = 1, 2, \ldots, \tau_2 + 1$, and both $z_0$ and $z_{\tau+1}$ are an empty sequence. Note that subsequence $S_2^i$ could be an empty sequence, for $i = 1, 2, \ldots, \tau_2 + 1$. We characterize the value of elements in $\mathcal{Z}_\tau^2(S)$ in the following:

$$\begin{aligned}
\eta \cdot h(S_2) &\overset{(a)}{=} h(S_2) - h(S_2 - \mathcal{Z}_\tau^2(S)) \\
&= h(S_2^1) + h((z_1)|S_2^1) + h(S_2^2|S_2^1 \oplus (z_1)) + \ldots \\
&\quad + h(S_2^{\tau_2+1}|S_2^1 \oplus (z_1) \oplus \cdots \oplus S_2^{\tau_2} \oplus (z_{\tau_2})) \\
&\quad - h(S_2^1) - h(S_2^2|S_2^1) - \cdots - h(S_2^{\tau_2+1}|S_2^1 \oplus \cdots \oplus S_2^{\tau_2}) \\
&= \sum_{i=1}^{\tau_2} h((z_i)|S_2^1 \oplus (z_1) \oplus \cdots \oplus (z_{i-1}) \oplus S_2^i) + h(S_2^2|S_2^1 \oplus (z_1)) - h(S_2^2|S_2^1) \\
&\quad + \cdots + h(S_2^{\tau_2+1}|S_2^1 \oplus (z_1) \oplus \cdots \oplus S_2^{\tau_2} \oplus (z_{\tau_2})) - h(S_2^{\tau_2+1}|S_2^1 \oplus \cdots \oplus S_2^{\tau_2}) \\
&\leq \sum_{i=1}^{\tau_2} h((z_i)|S_2^1 \oplus (z_1) \oplus \cdots \oplus (z_{i-1}) \oplus S_2^i) \\
&\leq \sum_{i=1}^{\tau_2} h((z_i)),
\end{aligned} \tag{52}$$

where (a) is from the definition of $\eta$ and the two inequalities are due to the general sequence submodularity of function $h$.

To prove Eq. (50a), we decompose it into two parts: (i) $h(S - \mathcal{Z}_\tau(S)) \geq \frac{\eta}{\tau} \cdot h(S_2)$ and (ii) $h(S - \mathcal{Z}_\tau(S)) \geq (1 - \eta) \cdot h(S_2)$.

Let $v'$ denote the first element in $S_1 - \mathcal{Z}_\tau^1(S)$. Then, Part (i) can be shown through the following:

$$h(S - \mathcal{Z}_\tau(S)) \overset{(a)}{\geq} h((v')) \overset{(b)}{\geq} \frac{1}{\tau_2} \sum_{i=1}^{\tau_2} h((z_i)) \overset{(c)}{\geq} \frac{\eta}{\tau_2} \cdot h(S_2) \overset{(d)}{\geq} \frac{\eta}{\tau} \cdot h(S_2),$$

where (a) is due to the forward monotonicity of function $h$, (b) is due to $h(v') \geq h(z_i)$ for any $i = 1, 2, \ldots, \tau_2$, (c) is from Eq. (52), and (d) is due to $\tau_2 \leq \tau$.

Part (ii) can be shown through the following:

$$h(S - \mathcal{Z}_\tau(S)) = h((S_1 - \mathcal{Z}_\tau^1(S)) \oplus (S_2 - \mathcal{Z}_\tau^2(S))) \overset{(a)}{\geq} h(S_2 - \mathcal{Z}_\tau^2(S)) \overset{(b)}{=} (1 - \eta) \cdot h(S_2),$$

where (a) is from the backward monotonicity of function $h$ and (b) is from the definition of $\eta$.

Eq. (50b) holds trivially for any $\eta \in (0, 1]$ by setting $\frac{\eta}{\tau}$ and $1 - \eta$ to be equal, solving for $\eta$, and plugging it back. This completes the proof. $\qquad\square$

## 8.8 Preliminary Results: Approximate Sequence Submodular Maximization

In this section, we generalize the results of Lemmas 1 and 2 under weaker assumptions based on approximate versions of sequence submodularity and backward monotonicity. Specifically, we introduce two weaker assumptions that will be used in this section.

**Assumption 6.** *Function $h$ is forward-monotone, $\alpha$-backward-monotone, and $\mu_1$-element-sequence-submodular.*

**Assumption 7.** *Function $h$ is forward-monotone, backward-monotone, and $\mu_1$-element-sequence-submodular.*

We first generalize the approximation result of the SSG algorithm (Algorithm 3) under Assumption 6. Let sequence $S$ with $|S| = k$ be the one selected by the SSG algorithm, let $S^i$ be the sequence consisting of the first $i$ elements of sequence $S$ with $1 \leq i \leq k$. We borrow the result of [28] and restate a generalized version in Lemma 6. This is also a generalization of Lemma 1. While [28] assumes that function $h$ is forward-monotone, $\alpha$-backward-monotone, and element-sequence-submodular and considers the case of $i = k$ only, the generalization to the result in Lemma 6 is fairly straightforward. Therefore, we refer the interested reader to [28] for the proof.

**Lemma 6** (Theorem 1 of [28]). *Consider $1 \leq i \leq k$. Under Assumption 6, we have $h(S^i) \geq \alpha(1 - \frac{1}{e^{\mu_1 \cdot \frac{i}{k}}})h(S^*(\mathcal{V}, k, 0))$.*

We also generalize the result of Lemma 2 and present the generalized result in Lemma 7. Note that the current proof techniques we use could only lead to the generalization under Assumption 7, which is slightly stronger than Assumption 6 (backward-monotone vs. $\alpha$-backward-monotone).

**Lemma 7.** *Consider $c \in (0, 1]$ and $1 \leq k' \leq k$. Suppose that the sequence selected by the SSG algorithm is $S$ with $|S| = k$ and that there exists a sequence $S_1$ with $|S_1| = k - k'$ such that $S_1 \preceq S$ and $h(S_1) \geq c \cdot h(S)$. Then, under Assumption 7, we have $h(S) \geq \frac{e^{\mu_1 \cdot \frac{k'}{k}} - 1}{e^{\mu_1 \cdot \frac{k'}{k}} - c} h(S^*(\mathcal{V}, k, 0))$.*

Before we prove Lemma 7, we introduce two lemmas: Lemmas 8 and 9. Lemma 8 is a generalization of Lemma 4 and will be used in the proof of Lemma 9, which will be used in the proof of Lemma 7.

**Lemma 8.** *Suppose that function $h$ is $\mu_1$-element-sequence-submodular. For any sequences $S_1', S_2' \in \mathcal{H}$, there exists an element $v \in \mathcal{V}$ such that $h((v)|S_1') \geq \frac{\mu_1}{|S_2'|}h(S_2'|S_1')$.*

*Proof.* Let $S_2' = (u_1, \ldots, u_{|S_2'|})$. We can rewrite $h(S_2'|S_1')$ as

$$h(S_2'|S_1') = \sum_{j=1}^{|S_2'|} h((u_j)|S_1' \oplus (u_1, \ldots, u_{j-1})), \qquad (53)$$

where sequence $(u_1, \ldots, u_{j-1})$ is an empty sequence when $j = 1$. Due to the Pigeonhole principle, there exists some $j' \in \{1, 2, \ldots |S'|\}$ such that the following is satisfied:

$$
\begin{aligned}
h((u_{j'})|S_1' \oplus (u_1, \ldots, u_{j'-1})) &\geq \frac{1}{|S_2'|} \sum_{j=1}^{|S_2'|} h((u_j)|S_1' \oplus (u_1, \ldots, u_{j-1})) \\
&= \frac{1}{|S_2'|} h(S_2'|S_1'),
\end{aligned}
\tag{54}
$$

where the equality is from Eq. (53). Furthermore, element $u_{j'}$ satisfies the following:

$$
\begin{aligned}
h((u_{j'})|S_1') &\overset{(a)}{\geq} \mu_1 \cdot h((u_{j'})|S_1' \oplus (u_1, \ldots, u_{j'-1})) \\
&\overset{(b)}{\geq} \frac{\mu_1}{|S_2'|} h(S_2'|S_1'),
\end{aligned}
\tag{55}
$$

where (a) is due to function $h$ being $\mu_1$-element-sequence-submodular and (b) is from Eq. (54). This completes the proof. $\qquad\square$

**Lemma 9.** *Consider $1 \leq k' \leq k$. Suppose that the sequence selected by the SSG algorithm is $S$ with $|S| = k$ and that there exist sequences $S_1$ and $S_2$ such that sequence $S$ can be written as $S = S_1 \oplus S_2$ with $|S_1| = k - k'$ and $|S_2| = k'$. Then, under Assumption 7, we have $h(S_2|S_1) \geq (1 - \frac{1}{e^{\mu_1 \cdot \frac{k'}{k}}})(h(S^*(\mathcal{V}, k, 0)) - h(S_1))$.*

*Proof.* The proof follows a similar line of analysis as in Lemma 5. Let $v_i$ denote the $i$-th element of sequence $S_2$, and let $S_2^i \triangleq (v_1, \ldots, v_i)$ denote the sequence consisting of the first $i$ elements of sequence $S_2$. Since function $h$ is forward-monotone, we can assume that $|S^*(\mathcal{V}, k, 0)| = k$ as adding more elements to the end of a sequence does not reduce its overall value.

Due to Lemma 8, there exists some element $v' \in \mathcal{V}$ such that $h((v')|S_1 \oplus S_2^{i-1}) \geq \frac{\mu_1}{k} h(S^*(\mathcal{V}, k, 0)|S_1 \oplus S_2^{i-1})$. Then, we have the following:

$$
\begin{aligned}
h(S_1 \oplus S_2^i) - h(S_1 \oplus S_2^{i-1}) &= h((v_i)|S_1 \oplus S_2^{i-1}) \\
&\overset{(a)}{\geq} h((v')|S_1 \oplus S_2^{i-1}) \\
&\overset{(b)}{\geq} \frac{\mu_1}{k} h(S^*(\mathcal{V}, k, 0)|S_1 \oplus S_2^{i-1}) \\
&= \frac{\mu_1}{k} (h(S_1 \oplus S_2^{i-1} \oplus S^*(\mathcal{V}, k, 0)) - h(S_1 \oplus S_2^{i-1})) \\
&\overset{(c)}{\geq} \frac{\mu_1}{k} (h(S^*(\mathcal{V}, k, 0)) - h(S_1 \oplus S_2^{i-1})),
\end{aligned}
\tag{56}
$$

where (a) is due to the greedy manner of the SSG algorithm (Line 4 of Algorithm 3), (b) is from the property of element $v'$ (due to Lemma 8), and (c) is due to the backward monotonicity of function $h$. Rewriting Eq. (56) yields the following equivalent inequality:

$$
h(S_1 \oplus S_2^i) \geq \frac{\mu_1}{k} h(S^*(\mathcal{V}, k, 0)) + (1 - \frac{\mu_1}{k}) h(S_1 \oplus S_2^{i-1}).
\tag{57}
$$

By writing Eq. (57) for $i \in \{1, \ldots, k'\}$ and combining them, we obtain the following:

$$
\begin{aligned}
h(S_1 \oplus S_2^{k'}) &\geq \sum_{j=0}^{k'-1} \frac{\mu_1}{k} (1 - \frac{\mu_1}{k})^j \, h(S^*(\mathcal{V}, k, 0)) + (1 - \frac{\mu_1}{k})^{k'} h(S_1) \\
&= (1 - (1 - \frac{\mu_1}{k})^{k'}) h(S^*(\mathcal{V}, k, 0)) + (1 - \frac{\mu_1}{k})^{k'} h(S_1).
\end{aligned}
\tag{58}
$$

Applying Eq. (58) and the fact that $S_2 = S_2^{k'}$ yields the following:

$$
\begin{aligned}
h(S_2|S_1) &= h(S_1 \oplus S_2^{k'}) - h(S_1) \\
&\geq (1 - (1 - \frac{\mu_1}{k})^{k'})h(S^*(\mathcal{V}, k, 0)) + (1 - \frac{\mu_1}{k})^{k'} h(S_1) - h(S_1) \\
&= (1 - (1 - \frac{\mu_1}{k})^{k'})h(S^*(\mathcal{V}, k, 0)) - (1 - (1 - \frac{\mu_1}{k})^{k'})h(S_1) \\
&= (1 - (1 - \frac{\mu_1}{k})^{k'})(h(S^*(\mathcal{V}, k, 0)) - h(S_1)) \\
&\geq (1 - \frac{1}{e^{\mu_1 \cdot \frac{k'}{k}}})(h(S^*(\mathcal{V}, k, 0)) - h(S_1)),
\end{aligned} \tag{59}
$$

where the last inequality holds because $(1 - \frac{\mu_1}{k}) \leq e^{-\frac{\mu_1}{k}}$ and $h(S^*(\mathcal{V}, k, 0)) - h(S_1)$ is nonnegative. This completes the proof. $\qquad\square$

Having introduced Lemmas 8 and 9, we are now ready to prove Lemma 7.

*Proof of Lemma 7.* Suppose $h(S) = \delta \cdot h(S^*(\mathcal{V}, k, 0))$ for some $\delta \in (0, 1]$. Then, we have

$$
\begin{aligned}
\delta \cdot h(S^*(\mathcal{V}, k, 0)) &= h(S) \\
&= h(S_1) + h(S_2|S_1) \\
&\overset{(a)}{\geq} h(S_1) + (1 - \frac{1}{e^{\mu_1 \cdot \frac{k'}{k}}})(h(S^*(\mathcal{V}, k, 0)) - h(S_1)) \\
&= \frac{1}{e^{\mu_1 \cdot \frac{k'}{k}}} h(S_1) + (1 - \frac{1}{e^{\mu_1 \cdot \frac{k'}{k}}})h(S^*(\mathcal{V}, k, 0)) \\
&\overset{(b)}{\geq} \frac{c\delta}{e^{\mu_1 \cdot \frac{k'}{k}}} h(S^*(\mathcal{V}, k, 0)) + (1 - \frac{1}{e^{\mu_1 \cdot \frac{k'}{k}}})h(S^*(\mathcal{V}, k, 0)),
\end{aligned} \tag{60}
$$

where (a) is from Lemma 9 and (b) is due to $h(S_1) \geq c \cdot h(S) = c \cdot \delta \cdot h(S^*(\mathcal{V}, k, 0))$. Dividing both sides of Eq. (60) by $h(S^*(\mathcal{V}, k, 0))$ yields the following:

$$
\delta \geq \frac{c\delta}{e^{\mu_1 \cdot \frac{k'}{k}}} + 1 - \frac{1}{e^{\mu_1 \cdot \frac{k'}{k}}}, \tag{61}
$$

which implies

$$
\delta \geq \frac{e^{\mu_1 \cdot \frac{k'}{k}} - 1}{e^{\mu_1 \cdot \frac{k'}{k}} - c}. \tag{62}
$$

Since $h(S) = \delta \cdot h(S^*(\mathcal{V}, k, 0))$, we have $h(S) \geq \frac{e^{\mu_1 \cdot \frac{k'}{k}} - 1}{e^{\mu_1 \cdot \frac{k'}{k}} - c} h(S^*(\mathcal{V}, k, 0))$. This completes the proof. $\qquad\square$

## 8.9  Proof of Theorem 4

*Proof.* We use Lemma 3 presented in Appendix 8.3 and Lemmas 6 and 7 presented in Appendix 8.8 to prove the approximation ratios of Algorithm 1 in the case of $\tau = 1$. The proof follows a similar line of analysis as in the proof of Theorem 1.

Given $\tau = 1$, in Step 1 of Algorithm 1, the selected sequence $S_1$ consists of one element only; this element is denoted by $v_1$, i.e., $S_1 = (v_1)$. In Step 2 of Algorithm 1, it is equivalent that sequence $S_2$ is selected by the SSG algorithm from set $\mathcal{V} \setminus \{v_1\}$, and we have $|S_2| = k - \tau = k - 1$. Hence, the sequence selected by Algorithm 1 can be written as $S = S_1 \oplus S_2 = (v_1) \oplus S_2$. Recall that for any given sequence $S$, set $\mathcal{Z}_\tau(S)$ denotes the set of elements removed from sequence $S$ in the worst case (i.e., $\mathcal{Z}_\tau(S)$ is an optimal solution to Problem (11)). Note that only one element will be removed from $S$, i.e., $|\mathcal{Z}_\tau(S)| = 1$. For ease of notation, we use $z$ to denote the only element in $\mathcal{Z}_\tau(S)$, i.e., $\mathcal{Z}_\tau(S) = \{z\}$.

We want to show the following two bounds, which establish the approximation ratios of Algorithm 1: 1) Suppose that function $h$ is forward-monotone, backward-monotone, $\mu_1$-element-sequence-submodular, and $\mu_2$-sequence-submodular (Assumption 3), we have

$$h(S - \{z\}) \geq \frac{a(e^b - 1)}{e^b - a} g_\tau(S^*(\mathcal{V}, k, \tau)), \text{ where } a = \frac{\mu_1 \mu_2}{\mu_1 + 1} \text{ and } b = \frac{\mu_1(k - 2)}{k - 1}; \qquad (63)$$

2) Suppose that function $h$ is forward-monotone, $\alpha$-backward-monotone, $\mu_1$-element-sequence-submodular, and $\mu_2$-sequence-submodular (Assumption 4), we have

$$h(S - \{z\}) \geq \frac{\alpha^2 \mu_1 \mu_2 (e^{\mu_1} - 1)}{(\mu_1 + \alpha)e^{\mu_1}} g_\tau(S^*(\mathcal{V}, k, \tau)). \qquad (64)$$

We begin with the proof of Eq. (64) as some of the intermediate results will be used in the proof of Eq. (63). We first present a lower bound on $h(S_2)$, which will be used throughout the proof:

$$\begin{aligned} h(S_2) &\geq \alpha(1 - 1/e^{\mu_1})h(S^*(\mathcal{V} \setminus \{v_1\}, k - \tau, 0)) \\ &\geq \alpha(1 - 1/e^{\mu_1})g_\tau(S^*(\mathcal{V}, k, \tau)), \end{aligned} \qquad (65)$$

where the first inequality is from Lemma 6 (where we replace $\mathcal{V}$, $k$, and $i$ with $\mathcal{V} \setminus \{v_1\}$, $k - \tau$, and $k - \tau$, respectively) and the second inequality is from Lemma 3 (where we replace $\mathcal{V}'$ with $\{v_1\}$).

The proof proceeds as follows. Element $z$ is an element that will be removed, which can be either $v_1$ or an element in $S_2$. Therefore, we consider two cases: (I) $z = v_1$ and (II) $z \neq v_1$.

In Case I, we have $z = v_1$, which implies the following:

$$h(S - \{z\}) = h(S_2) \geq \alpha(1 - 1/e^{\mu_1})g_\tau(S^*(\mathcal{V}, k, \tau)), \qquad (66)$$

where the inequality follows from Eq. (65).

In Case II, we have $z \neq v_1$ (or $z \in \mathcal{V}(S_2)$). Depending on the impact of removing element $z$, we consider two subcases: (II-a) $h(S_2) \leq h(S_2 - \{z\})$ and (II-b) $h(S_2) > h(S_2 - \{z\})$.

In Case II-a: we have $h(S_2) \leq h(S_2 - \{z\})$. In this case, the removal of element $z$ does not reduce the overall value of the remaining sequence $S_2 - \{z\}$. Then, we have

$$\begin{aligned} h(S - \{z\}) &= h((v_1) \oplus (S_2 - \{z\})) \\ &\overset{(a)}{\geq} \alpha \cdot h(S_2 - \{z\}) \\ &\overset{(b)}{\geq} \alpha \cdot h(S_2) \\ &\overset{(c)}{\geq} \alpha^2 \cdot (1 - 1/e^{\mu_1})g_\tau(S^*(\mathcal{V}, k, \tau)), \end{aligned} \qquad (67)$$

where (a) is due to the $\alpha$-backward monotonicity of function $h$, (b) holds from the condition of this subcase, and (c) follows from Eq. (65).

In Case II-b: we have $h(S_2) > h(S_2 - \{z\})$. Let $\eta \triangleq \frac{h(S_2) - h(S_2 - \{z\})}{h(S_2)}$ denote the ratio of the loss of removing element $z$ from sequence $S_2$ to the value of sequence $S_2$, and we have $\eta \in (0, 1]$ due to $h(S_2) > h(S_2 - \{z\})$. We first state the following:

$$h(S - \{z\}) \geq \max\{\mu_1(\eta + \mu_2 - 1) \cdot h(S_2), \alpha(1 - \eta) \cdot h(S_2)\}, \qquad (68a)$$

$$\max\{\mu_1(\eta + \mu_2 - 1), \alpha(1 - \eta)\} \geq \frac{\alpha \mu_1 \mu_2}{\mu_1 + \alpha}. \qquad (68b)$$

We will prove Eqs. (68a) and (68b) later; for now, we assume that they both hold. Then, we can obtain the following bound:

$$\begin{aligned} h(S - \{z\}) &\geq \max\{\mu_1(\eta + \mu_2 - 1) \cdot h(S_2), \alpha(1 - \eta) \cdot h(S_2)\} \\ &\geq \max\{\mu_1(\eta + \mu_2 - 1), \alpha(1 - \eta)\} \cdot \alpha(1 - 1/e^{\mu_1})g_\tau(S^*(\mathcal{V}, k, \tau)) \\ &\geq \frac{\alpha^2 \mu_1 \mu_2 (e^{\mu_1} - 1)}{(\mu_1 + \alpha)e^{\mu_1}} g_\tau(S^*(\mathcal{V}, k, \tau)), \end{aligned} \qquad (69)$$

where the three inequalities are from Eqs. (68a), (65), and (68b), respectively.

By combining the bounds in Eqs. (66), (67), and (69), we obtain the bound in Eq. (64).

Next, we prove Eq. (68a). We first rewrite $S_2$ as $S_2 = S_2^1 \oplus (z) \oplus S_2^2$, where $S_2^1$ and $S_2^2$ denote the subsequences of $S_2$ before and after element $z$, respectively. Note that $S_2^1$ or $S_2^2$ could be an empty sequence, depending on the position of $z$ in $S_2$. Then, we characterize $h((z))$ in terms of $h(S_2)$:

$$
\begin{aligned}
\eta \cdot h(S_2) &\overset{(a)}{=} h(S_2) - h(S_2 - \{z\}) \\
&= h(S_2^1 \oplus (z) \oplus S_2^2) - h(S_2^1 \oplus S_2^2) \\
&= h(S_2^1) + h((z)|S_2^1) + h(S_2^2|S_2^1 \oplus (z)) - h(S_2^1) - h(S_2^2|S_2^1) \\
&= h((z)|S_2^1) + h(S_2^2|S_2^1 \oplus (z)) - h(S_2^2|S_2^1) \\
&\overset{(b)}{\leq} h((z)|S_2^1) + h(S_2^2|S_2^1 \oplus (z)) - \mu_2 h(S_2^2|S_2^1 \oplus (z)) \\
&= h((z)|S_2^1) + (1 - \mu_2) h(S_2^2|S_2^1 \oplus (z)) \\
&\overset{(c)}{\leq} h((z)|S_2^1) + (1 - \mu_2) h(S_2) \\
&\overset{(d)}{\leq} h((z))/\mu_1 + (1 - \mu_2) h(S_2),
\end{aligned}
\tag{70}
$$

where (a) is from the definition of $\eta$, (b) is due to function $h$ being $\mu_2$-sequence-submodular, (c) holds because $\mu_2 \leq 1$ and $h(S_2^2|S_2^1 \oplus (z)) \leq h(S_2^2|S_2^1 \oplus (z)) + h(S_2^1 \oplus (z)) = h(S_2)$ (recall that $S_2 = S_2^1 \oplus (z) \oplus S_2^2$), and (d) is due to function $h$ being $\mu_1$-element-sequence-submodular. From Eq. (70), we have the following:

$$
h((z)) \geq \mu_1 (\eta + \mu_2 - 1) h(S_2). \tag{71}
$$

To prove Eq. (68a), we decompose it into two parts: (i) $h(S - \{z\}) \geq \mu_1 (\eta + \mu_2 - 1) \cdot h(S_2)$ and (ii) $h(S - \{z\}) \geq \alpha (1 - \eta) \cdot h(S_2)$.

Part (i) can be shown through the following:

$$
h(S - \{z\}) \overset{(a)}{\geq} h((v_1)) \overset{(b)}{\geq} h((z)) \overset{(c)}{\geq} \mu_1 (\eta + \mu_2 - 1) \cdot h(S_2), \tag{72}
$$

where (a) is from the forward monotonicity of function $h$, (b) is due to the greedy manner of Algorithm 1 (Lines 3-5), and (c) is from Eq. (71).

Part (ii) can be shown through the following:

$$
h(S - \{z\}) = h((v_1) \oplus (S_2 - \{z\})) \overset{(a)}{\geq} \alpha \cdot h(S_2 - \{z\}) \overset{(b)}{=} \alpha (1 - \eta) \cdot h(S_2),
$$

where (a) is from the $\alpha$-backward monotonicity of function $h$ and (b) is from the definition of $\eta$.

Eq. (68b) holds trivially for any $\eta \in (0, 1]$ by setting $\mu_1 (\eta + \mu_2 - 1)$ and $\alpha (1 - \eta)$ to be equal, solving for $\eta$, and plugging it back. This completes the proof of the bound in Eq. (64).

Next, we prove the bound in Eq. (63). Note that the analysis so far applies to function $h$ that is forward-monotone, $\alpha$-backward-monotone, $\mu_1$-element-sequence-submodular, and $\mu_2$-sequence-submodular (Assumption 4), and hence, it also applies to Assumption 3, which is a special case of Assumption 4 with $\alpha = 1$. The bound in Eq. (63) requires backward monotonicity (i.e., $\alpha = 1$), but it becomes better than the bound in Eq. (64) when $k$ is large. In the following analysis, we assume that function $h$ is forward-monotone, backward-monotone, $\mu_1$-element-sequence-submodular, and $\mu_2$-sequence-submodular (Assumption 3).

The proof proceeds as follows. We borrow the analysis of Case-I, Case II-a, and Eq. (68a) from the previous analysis by setting $\alpha = 1$. For Case-II-b, we provide a different analysis. In Case II-b: we have $z \in \mathcal{V}(S_2)$ and $h(S_2) > h(S_2 - \{z\})$. Suppose $k = 2$. Then, it is trivial that the sequence selected by Algorithm 1 (i.e., $S = (v_1) \oplus (z)$) yields an optimal solution. This is because removing element $z$ from $S$ gives $(v_1)$, which has the largest individual value among all the elements. Therefore, we assume $k > 2$ throughout the rest of the proof. Recall that $\eta \triangleq \frac{h(S_2) - h(S_2 - \{z\})}{h(S_2)}$. We

first state the following:

$$h(S_2) \geq \frac{e^{\mu_1 \cdot \frac{k-2}{k-1}} - 1}{e^{\mu_1 \cdot \frac{k-2}{k-1}} - \mu_1(\eta + \mu_2 - 1)} g_\tau(S^*(\mathcal{V}, k, \tau)), \tag{73a}$$

$$\max\left\{ \frac{\mu_1(\eta + \mu_2 - 1)(e^{\mu_1 \cdot \frac{k-2}{k-1}} - 1)}{e^{\mu_1 \cdot \frac{k-2}{k-1}} - \mu_1(\eta + \mu_2 - 1)}, \frac{(1 - \eta)(e^{\mu_1 \cdot \frac{k-2}{k-1}} - 1)}{e^{\mu_1 \cdot \frac{k-2}{k-1}} - \mu_1(\eta + \mu_2 - 1)} \right\}$$

$$\geq \frac{\mu_1 \mu_2}{1 + \mu_2} \cdot \frac{e^{\mu_1 \cdot \frac{k-2}{k-1}} - 1}{e^{\mu_1 \cdot \frac{k-2}{k-1}} - \frac{\mu_1 \mu_2}{1 + \mu_2}}. \tag{73b}$$

We will prove Eqs. (73a) and (73b) later; for now, we assume that they both hold. Then, we can obtain the following bound:

$$h(S - \{z\}) \geq \max\{\mu_1(\eta + \mu_2 - 1) \cdot h(S_2), (1 - \eta) \cdot h(S_2)\}$$

$$\geq \max\left\{ \frac{\mu_1(\eta + \mu_2 - 1)(e^{\mu_1 \cdot \frac{k-2}{k-1}} - 1)}{e^{\mu_1 \cdot \frac{k-2}{k-1}} - \mu_1(\eta + \mu_2 - 1)}, \frac{(1 - \eta)(e^{\mu_1 \cdot \frac{k-2}{k-1}} - 1)}{e^{\mu_1 \cdot \frac{k-2}{k-1}} - \mu_1(\eta + \mu_2 - 1)} \right\} g_\tau(S^*(\mathcal{V}, k, \tau))$$

$$\geq \frac{\mu_1 \mu_2}{1 + \mu_2} \cdot \frac{e^{\mu_1 \cdot \frac{k-2}{k-1}} - 1}{e^{\mu_1 \cdot \frac{k-2}{k-1}} - \frac{\mu_1 \mu_2}{1 + \mu_2}} \cdot g_\tau(S^*(\mathcal{V}, k, \tau)),$$

$$\tag{74}$$

where the three inequalities are from Eq. (68a) (with $\alpha = 1$), (73a), and (73b), respectively.

By combining the bounds in Eqs. (66), (67) (with $\alpha = 1$), and (74), we obtain Eq. (63).

Next, we show that Eq. (73a) holds. Let $v_2^1$ denote the first element of sequence $S_2$. Then, we have the following:

$$h((v_2^1)) \overset{\text{(a)}}{\geq} h((z)) \overset{\text{(b)}}{\geq} \mu_1(\eta + \mu_2 - 1) \cdot h(S_2),$$

where (a) holds because element $v_2^1$ has the largest individual value among all elements in $S_2$ and (b) follows from Eq. (71).Then, we can characterize the value of $h(S_2)$ as follows:

$$h(S_2) \geq \frac{e^{\mu_1 \cdot \frac{k-2}{k-1}} - 1}{e^{\mu_1 \cdot \frac{k-2}{k-1}} - \mu_1(\eta + \mu_2 - 1)} h(S^*(\mathcal{V} \setminus \{v_1\}, k - \tau, 0))$$

$$\geq \frac{e^{\mu_1 \cdot \frac{k-2}{k-1}} - 1}{e^{\mu_1 \cdot \frac{k-2}{k-1}} - \mu_1(\eta + \mu_2 - 1)} g_\tau(S^*(\mathcal{V}, k, \tau)),$$

where the inequalities are from Lemma 7 (where we replace $\mathcal{V}$, $S$, $S_1$, $k$, $k'$, and $c$ with $\mathcal{V} \setminus \{v_1\}$, $S_2$, $(v_2^1)$, $k - 1$, $k - 2$, and $\mu_1(\eta + \mu_2 - 1)$, respectively) and Lemma 3, respectively.

Finally, we show that Eq. (73b) holds. We define two functions of $\eta$: $l_1(\eta) \triangleq \frac{\mu_1(\eta + \mu_2 - 1)(e^{\mu_1 \cdot \frac{k-2}{k-1}} - 1)}{e^{\mu_1 \cdot \frac{k-2}{k-1}} - \mu_1(\eta + \mu_2 - 1)}$ and $l_2(\eta) \triangleq \frac{(1 - \eta)(e^{\mu_1 \cdot \frac{k-2}{k-1}} - 1)}{e^{\mu_1 \cdot \frac{k-2}{k-1}} - \mu_1(\eta + \mu_2 - 1)}$. It is easy to verify that for $k > 2$ and $\eta \in (0, 1]$, function $l_1(\eta)$ is monotonically increasing and function $l_2(\eta)$ is monotonically decreasing. Let $\eta^* = \frac{1 + \mu_1 - \mu_1 \mu_2}{1 + \mu_1}$, and we have $l_1(\eta^*) = l_2(\eta^*) = \frac{\mu_1 \mu_2}{1 + \mu_2} \cdot \frac{e^{\mu_1 \cdot \frac{k-2}{k-1}} - 1}{e^{\mu_1 \cdot \frac{k-2}{k-1}} - \frac{\mu_1 \mu_2}{1 + \mu_2}}$. We consider two cases for $\eta$: $\eta \in [\eta^*, 1]$ and $\eta \in (0, \eta^*]$. For $\eta \in [\eta^*, 1]$, we have $\max\{l_1(\eta), l_2(\eta)\} \geq l_1(\eta) \geq l_1(\eta^*)$ as $l_1(\eta)$ is monotonically increasing; for $\eta \in (0, \eta^*]$, we have $\max\{l_1(\eta), l_2(\eta)\} \geq l_2(\eta) \geq l_2(\eta^*) = l_1(\eta^*)$ as $l_2(\eta)$ is monotonically decreasing. Therefore, for $\eta \in (0, 1]$, we have $\max\{l_1(\eta), l_2(\eta)\} \geq l_1(\eta^*) = \frac{\mu_1 \mu_2}{1 + \mu_2} \cdot \frac{e^{\mu_1 \cdot \frac{k-2}{k-1}} - 1}{e^{\mu_1 \cdot \frac{k-2}{k-1}} - \frac{\mu_1 \mu_2}{1 + \mu_2}}$. This gives Eq. (73b) and completes the proof. $\square$

## 8.10 Proof of Theorem 5

*Proof.* We use Lemma 3 presented in Appendix 8.3 and Lemmas 6 and 7 presented in Appendix 8.8 to prove the approximation ratios of Algorithm 1 in the case of $1 \leq \tau \leq k$, assuming the removal of $\tau$ contiguous elements. The proof follows a similar line of analysis as in the proof of Theorem 2.

In Step 1 of Algorithm 1, it is equivalent that sequence $S_1$ is selected by the SSG algorithm from set $\mathcal{V}$, and we have $|S_1| = \tau$. Similarly, in Step 2 of Algorithm 1, it is also equivalent that sequence $S_2$ is selected by the SSG algorithm from set $\mathcal{V} \setminus \mathcal{V}(S_1)$, and we have $|S_2| = k - \tau$. Hence, the sequence selected by Algorithm 1 can be written as $S = S_1 \oplus S_2$. Recall that for any given sequence $S$, set $\mathcal{Z}_\tau(S)$ denotes the set of elements removed from sequence $S$ in the worst case (i.e., $\mathcal{Z}_\tau(S)$ is an optimal solution to Problem (11)). We define $\mathcal{Z}_\tau^1(S) \triangleq \mathcal{Z}_\tau(S) \cap \mathcal{V}(S_1)$ and $\mathcal{Z}_\tau^2(S) \triangleq \mathcal{Z}_\tau(S) \cap \mathcal{V}(S_2)$ as the set of elements removed from subsequences $S_1$ and $S_2$, respectively.

We want to show the following two bounds, which establish the approximation ratios of Algorithm 1: 1) Suppose that function $h$ is forward-monotone, backward-monotone, $\mu_1$-element-sequence-submodular, and $\mu_2$-sequence-submodular (Assumption 3), we have

$$h(S - \mathcal{Z}_\tau(S)) \geq \frac{a\mu_2(e^b - 1)}{(a+1)e^b - a\mu_2} g_\tau(S^*(\mathcal{V}, k, \tau)), \tag{75}$$

where $a = \mu_1 \cdot (1 - 1/e^{\mu_1})$ and $b = \mu_1 \cdot \frac{k-2\tau}{k-\tau}$; 2) Suppose that function $h$ is forward-monotone, $\alpha$-backward-monotone, $\mu_1$-element-sequence-submodular, and $\mu_2$-sequence-submodular (Assumption 4), we have

$$h(S - \mathcal{Z}_\tau(S)) \geq \frac{\alpha^2 \mu_1 \mu_2 (e^{\mu_1} - 1)^2}{\mu_1 e^{\mu_1}(e^{\mu_1} - 1) + e^{2\mu_1}} g_\tau(S^*(\mathcal{V}, k, \tau)). \tag{76}$$

We begin with the proof of Eq. (76) as some of the intermediate results will be used in the proof of Eq. (75). We first present a lower bound on $h(S_2)$, which will be used throughout the proof:

$$\begin{aligned}
h(S_2) &\geq \alpha(1 - 1/e^{\mu_1})h(S^*(\mathcal{V} \setminus \mathcal{V}(S_1), k - \tau, 0)) \\
&\geq \alpha(1 - 1/e^{\mu_1})g_\tau(S^*(\mathcal{V}, k, \tau)),
\end{aligned} \tag{77}$$

where the first inequality is from Lemma 6 (where we replace $\mathcal{V}$, $k$, and $i$ with $\mathcal{V} \setminus \mathcal{V}(S_1)$, $k - \tau$, and $k - \tau$, respectively) and the second inequality is from Lemma 3 (where we replace $\mathcal{V}'$ with $\mathcal{V}(S_1)$).

The proof proceeds as follows. Elements in $\mathcal{Z}_\tau(S)$ will be removed from sequence $S$. These elements can be either fully or partially in $\mathcal{V}(S_1)$ (i.e., $\mathcal{Z}_\tau^2(S) = \emptyset$ or $\mathcal{Z}_\tau^2(S) \neq \emptyset$). Therefore, we consider two cases: (I) $\mathcal{Z}_\tau^2(S) = \emptyset$ and (II) $\mathcal{Z}_\tau^2(S) \neq \emptyset$.

In Case I, we have $\mathcal{Z}_\tau^2(S) = \emptyset$, i.e., $\mathcal{Z}_\tau(S) = \mathcal{V}(S_1)$. Then, we have the following:

$$h(S - \mathcal{Z}_\tau(S)) = h(S_2) \geq \alpha(1 - 1/e^{\mu_1})g_\tau(S^*(\mathcal{V}, k, \tau)), \tag{78}$$

where the inequality follows from Eq. (77).

In Case II, we have $\mathcal{Z}_\tau^2(S) \neq \emptyset$. Depending on the impact of removing elements in $\mathcal{Z}_\tau^2(S)$, we consider two subcases: (II-a) $h(S_2) \leq h(S_2 - \mathcal{Z}_\tau^2(S))$ and (II-b) $h(S_2) > h(S_2 - \mathcal{Z}_\tau^2(S))$.

In Case II-a: we have $h(S_2) \leq h(S_2 - \mathcal{Z}_\tau^2(S))$. In this case, the removal of elements in $\mathcal{Z}_\tau^2(S)$ does not reduce the overall value of the remaining sequence $S_2 - \mathcal{Z}_\tau^2(S)$. Then, we have

$$\begin{aligned}
h(S - \mathcal{Z}_\tau(S)) &= h((S_1 - \mathcal{Z}_\tau^1(S)) \oplus (S_2 - \mathcal{Z}_\tau^2(S))) \\
&\overset{(a)}{\geq} \alpha \cdot h(S_2 - \mathcal{Z}_\tau^2(S)) \\
&\overset{(b)}{\geq} \alpha \cdot h(S_2) \\
&\overset{(c)}{\geq} \alpha^2(1 - 1/e^{\mu_1}) \cdot g_\tau(S^*(\mathcal{V}, k, \tau)),
\end{aligned} \tag{79}$$

where (a) is due to the $\alpha$-backward monotonicity of function $h$, (b) holds from the condition of this subcase, and (c) follows from Eq. (77).

In Case II-b: we have $h(S_2) > h(S_2 - \mathcal{Z}_\tau^2(S))$. Let $\eta \triangleq \frac{h(S_2) - h(S_2 - \mathcal{Z}_\tau^2(S))}{h(S_2)}$ denote the ratio of the loss of removing elements in $\mathcal{Z}_\tau^2(S)$ from sequence $S_2$ to the value of sequence $S_2$, and we have $\eta \in (0, 1]$ due to $h(S_2) > h(S_2 - \mathcal{Z}_\tau^2(S))$. We first state the following:

$$h(S - \mathcal{Z}_\tau(S)) \geq \max\{\mu_1\alpha(\eta + \mu_2 - 1)(1 - 1/e^{\mu_1}) \cdot h(S_2), \alpha(1 - \eta) \cdot h(S_2)\}, \tag{80a}$$

$$\max\{\mu_1\alpha(\eta + \mu_2 - 1)(1 - 1/e^{\mu_1}), \alpha(1 - \eta)\} \geq \frac{\alpha\mu_1\mu_2(e^{\mu_1} - 1)}{\mu_1(e^{\mu_1} - 1) + e^{\mu_1}}. \tag{80b}$$

We will prove Eqs. (80a) and (80b) later; for now, we assume that they both hold. Then, we can obtain the following bound:

$$
\begin{aligned}
h(S - \mathcal{Z}_\tau(S)) &\geq \max\{\mu_1 \alpha(\eta + \mu_2 - 1)(1 - 1/e^{\mu_1}) \cdot h(S_2), \alpha(1 - \eta) \cdot h(S_2)\} \\
&\geq \max\{\mu_1 \alpha(\eta + \mu_2 - 1)(1 - 1/e^{\mu_1}), \alpha(1 - \eta)\} \cdot \alpha(1 - 1/e^{\mu_1}) \cdot g_\tau(S^*(\mathcal{V}, k, \tau)) \\
&\geq \frac{\alpha^2 \mu_1 \mu_2 (e^{\mu_1} - 1)^2}{\mu_1 e^{\mu_1}(e^{\mu_1} - 1) + e^{2\mu_1}} \cdot g_\tau(S^*(\mathcal{V}, k, \tau)),
\end{aligned}
$$

(81)

where the three inequalities are from Eqs. (80a), (77), and (80b), respectively.

By combining the bounds in Eqs. (78), (79), and (81), we obtain the second bound in Eq. (76).

Next, we show that Eq. (80a) holds. Recall that the $\tau$ elements in $\mathcal{Z}_\tau(S)$ form a contiguous subsequence of $S$. Then, the elements in $\mathcal{Z}_\tau^1(S)$ and $\mathcal{Z}_\tau^2(S)$ also form a contiguous subsequence of $S_1$ and $S_2$, respectively. We use $Z_1$ and $Z_2$ to denote the contiguous subsequence of elements in $\mathcal{Z}_\tau^1(S)$ and $\mathcal{Z}_\tau^2(S)$, respectively. We first rewrite $S_2$ as $S_2 = S_2^1 \oplus Z_2 \oplus S_2^2$, where $S_2^1$ and $S_2^1$ denote the subsequences of $S_2$ before and after subsequence $Z_2$, respectively. Note that $S_2^1$ or $S_2^2$ could be an empty sequence, depending on the position of subsequence $Z_2$ in $S_2$. Then, we characterize $h(Z_2)$ in terms of $h(S_2)$:

$$
\begin{aligned}
\eta \cdot h(S_2) &\overset{(a)}{=} h(S_2) - h(S_2 - \mathcal{Z}_\tau^2(S)) \\
&= h(S_2^1 \oplus Z_2 \oplus S_2^2) - h(S_2^1 \oplus S_2^2) \\
&= h(S_2^1) + h(Z_2|S_2^1) + h(S_2^2|S_2^1 \oplus Z_2) - h(S_2^1) - h(S_2^2|S_2^1) \\
&= h(Z_2|S_2^1) + h(S_2^2|S_2^1 \oplus Z_2) - h(S_2^2|S_2^1) \\
&\overset{(b)}{\leq} h(Z_2|S_2^1) + h(S_2^2|S_2^1 \oplus Z_2) - \mu_2 h(S_2^2|S_2^1 \oplus Z_2) \\
&= h(Z_2|S_2^1) + (1 - \mu_2)h(S_2^2|S_2^1 \oplus Z_2) \\
&\overset{(c)}{\leq} h(Z_2|S_2^1) + (1 - \mu_2)h(S_2) \\
&\overset{(d)}{\leq} h(Z_2)/\mu_1 + (1 - \mu_2)h(S_2),
\end{aligned}
$$

(82)

where (a) is from the definition of $\eta$, (b) is due to function $h$ being $\mu_2$-sequence-submodular, (c) holds because $\mu_2 \leq 1$ and $h(S_2^2|S_2^1 \oplus Z_2) \leq h(S_2^2|S_2^1 \oplus Z_2) + h(S_2^1 \oplus Z_2) = h(S_2)$ (recall that $S_2 = S_2^1 \oplus Z_2 \oplus S_2^2$), and (d) is due to function $h$ being $\mu_1$-element-sequence-submodular. From Eq. (82), we have the following:

$$
h(Z_2) \geq \mu_1(\eta + \mu_2 - 1)h(S_2).
$$

(83)

To prove Eq. (80a), we decompose it into two parts: (i) $h(S - \mathcal{Z}_\tau(S)) \geq \mu_1 \alpha(\eta + \mu_2 - 1)(1 - 1/e^{\mu_1}) \cdot h(S_2)$ and (ii) $h(S - \mathcal{Z}_\tau(S)) \geq \alpha(1 - \eta) \cdot h(S_2)$.

Let $\tau_2 \triangleq |\mathcal{Z}_\tau^2(S)|$. We use $S_1^{\tau_2}$ to denote the subsequence consisting of the first $\tau_2$ elements in $S_1$. Then, Part (i) can be shown through the following:

$$
\begin{aligned}
h(S - \mathcal{Z}_\tau(S)) &= h((S_1 - \mathcal{Z}_\tau^1(S)) \oplus (S_2 - \mathcal{Z}_\tau^2(S))) \\
&\overset{(a)}{\geq} h(S_1 - \mathcal{Z}_\tau^1(S)) \\
&= h(S_1^{\tau_2}) \\
&\overset{(b)}{\geq} \alpha(1 - 1/e^{\mu_1}) \cdot h(S^*(\mathcal{V}, \tau_2, 0)) \\
&\overset{(c)}{\geq} \alpha(1 - 1/e^{\mu_1}) \cdot h(Z_2) \\
&\overset{(d)}{\geq} \mu_1 \alpha(\eta + \mu_2 - 1)(1 - 1/e^{\mu_1}) \cdot h(S_2),
\end{aligned}
$$

(84)

where (a) is due to the forward monotonicity of function $h$, (b) is due to the greedy manner of selecting subsequence $S_1^{\tau_2}$ from set $\mathcal{V}$ and Lemma 6 (where we replace both $k$ and $i$ with $\tau_2$), (c) holds because sequence $Z_2$ is a feasible solution to Problem $(P)$ for selecting a sequence of $\tau_2$ elements from $\mathcal{V}$, and (d) is from Eq. (83).

Part (ii) can be shown through the following:

$$h(S - \mathcal{Z}_\tau(S)) = h((S_1 - \mathcal{Z}_\tau^1(S)) \oplus (S_2 - \mathcal{Z}_\tau^2(S))) \overset{(a)}{\geq} \alpha \cdot h(S_2 - \mathcal{Z}_\tau^2(S)) \overset{(b)}{=} \alpha(1-\eta) \cdot h(S_2), \quad (85)$$

where (a) is from the $\alpha$-backward monotonicity of function $h$ and (b) is from the definition of $\eta$.

Eq. (80b) holds trivially for any $\eta \in (0, 1]$ by setting $\mu_1 \alpha(\eta + \mu_2 - 1)(1 - 1/e^{\mu_1})$ and $\alpha(1-\eta)$ to be equal, solving for $\eta$, and plugging it back. This completes the proof of the bound in Eq. (76).

Now, we prove the bound in Eq. (75). Note that the analysis so far applies to any function $h$ that is forward-monotone, $\alpha$-backward-monotone, $\mu_1$-element-sequence-submodular, and $\mu_2$-sequence-submodular (Assumption 4), and hence, it also applies to Assumption 3, which is a special case of Assumption 4 with $\alpha = 1$. The bound in Eq. (75) requires backward monotonicity (i.e., $\alpha = 1$), but it becomes better than the bound in Eq. (76) when $k$ is large. In the following analysis, we assume that function $h$ is forward-monotone, backward-monotone, $\mu_1$-element-sequence-submodular, and $\mu_2$-sequence-submodular (Assumption 3).

The proof proceeds as follows. We borrow the analysis of Case-I, Case II-a, and Eq. (80a) from the previous analysis by setting $\alpha = 1$. Then, for Case-II-b, we provide a different analysis.

In Case II-b: we have $h(S_2) > h(S_2 - \mathcal{Z}_\tau^2(S))$. Let $\tau_1 \triangleq |\mathcal{Z}_\tau^1(S)|$ and recall that $\tau_2 \triangleq |\mathcal{Z}_\tau^2(S)|$. Then, we have $\tau = \tau_1 + \tau_2$ and $k = |S| = |S_1| + |S_2| \geq \tau + \tau_2$. Therefore, we consider two cases: $k = \tau + \tau_2$ and $k > \tau + \tau_2$.

Suppose $k = \tau + \tau_2$. Then, it implies $\mathcal{Z}_\tau^2(S) = \mathcal{V}(S_2)$, i.e., all the elements in $S_2$ are removed. This further implies that the elements in $\mathcal{Z}_\tau^1(S)$ are at the end of sequence $S_1$ (due to the contiguous assumption of elements in $\mathcal{Z}_\tau(S)$). Let $S_1^{\tau_2} \preceq S_1$ denote the subsequence consisting of the first $\tau_2$ elements in $S_1$. It is easy to see $S_1 - \mathcal{Z}_\tau^1(S) = S_1^{\tau_2}$. Then, we have the following:

$$\begin{aligned}
h(S - \mathcal{Z}_\tau(S)) &= h((S_1 - \mathcal{Z}_\tau^1(S)) \oplus (S_2 - \mathcal{Z}_\tau^2(S))) \\
&= h(S_1 - \mathcal{Z}_\tau^1(S)) \\
&= h(S_1^{\tau_2}) \\
&\geq (1 - 1/e^{\mu_1})h(S^*(\mathcal{V}, \tau_2, 0)) \\
&\geq (1 - 1/e^{\mu_1})g_\tau(S^*(\mathcal{V}, k, \tau)),
\end{aligned} \quad (86)$$

where the first inequality is from Lemma 6 (where we replace both $k$ and $i$ with $\tau_2$ and set $\alpha = 1$) and the second inequality is due to $\tau_2 = k - \tau$ and Lemma 3 (where $\mathcal{V}'$ is an empty set).

Now, suppose $k > \tau + \tau_2$. Recall that $\eta \triangleq \frac{h(S_2) - h(S_2 - \mathcal{Z}_\tau^2(S))}{h(S_2)}$. Let $a \triangleq \mu_1(1 - 1/e^{\mu_1})$, $b \triangleq \mu_1 \cdot \frac{k-2\tau}{k-\tau}$, and $b' \triangleq \mu_1 \cdot \frac{k-\tau-\tau_2}{k-\tau_2}$. We first state the following:

$$h(S_2) \geq \frac{e^{b'} - 1}{e^{b'} - a(\eta + \mu_2 - 1)} g_\tau(S^*(\mathcal{V}, k, \tau)), \quad (87a)$$

$$\max\left\{ \frac{a(\eta + \mu_2 - 1)(e^{b'} - 1)}{e^{b'} - a(\eta + \mu_2 - 1)}, \frac{(1-\eta)(e^{b'} - 1)}{e^{b'} - a(\eta + \mu_2 - 1)} \right\} \geq \frac{a\mu_2(e^{b'} - 1)}{(a+1)e^{b'} - a\mu_2}. \quad (87b)$$

We will prove Eqs. (87a) and (87b) later; for now, we assume that they both hold. Then, we can obtain the following bound:

$$\begin{aligned}
h(S - \mathcal{Z}_\tau(S)) &\geq \max\{\mu_1(\eta + \mu_2 - 1)(1 - 1/e^{\mu_1}) \cdot h(S_2), (1-\eta) \cdot h(S_2)\} \\
&\geq \max\left\{ \frac{a(\eta + \mu_2 - 1)(e^{b'} - 1)}{e^{b'} - a(\eta + \mu_2 - 1)}, \frac{(1-\eta)(e^{b'} - 1)}{e^{b'} - a(\eta + \mu_2 - 1)} \right\} g_\tau(S^*(\mathcal{V}, k, \tau)) \\
&\geq \frac{a\mu_2(e^{b'} - 1)}{(a+1)e^{b'} - a\mu_2} g_\tau(S^*(\mathcal{V}, k, \tau)) \\
&\geq \frac{a\mu_2(e^b - 1)}{(a+1)e^b - a\mu_2} g_\tau(S^*(\mathcal{V}, k, \tau)),
\end{aligned} \quad (88)$$

where the first three inequalities are from Eq. (80a) ($\alpha = 1$) and Eqs. (87a) and (87b), respectively; for the last inequality, we replace $\tau_2$ in $b'$ with $\tau$ to obtain $b$, and the inequality holds due to $\tau_2 \leq \tau$ and that $\frac{a\mu_2(e^{b'} - 1)}{(a+1)e^{b'} - a\mu_2}$ is a decreasing function of $\tau_2$.

By combining the bounds in Eqs. (78), (79) (with $\alpha = 1$), (86), and (88), we obtain Eq. (75).

Next, we show that Eq. (87a) holds. Let $S_2^{\tau_2} \preceq S_2$ denote the subsequence consisting of the first $\tau_2$ elements of sequence $S_2$. Then, we have the following:

$$
\begin{aligned}
h(S_2^{\tau_2}) &\overset{(a)}{\geq} (1 - 1/e^{\mu_1}) h(S^*(\mathcal{V} \setminus \mathcal{V}(S_1), \tau_2, 0)) \\
&\overset{(b)}{\geq} (1 - 1/e^{\mu_1}) h(Z_2) \\
&\overset{(c)}{\geq} \mu_1(\eta + \mu_2 - 1)(1 - 1/e^{\mu_1}) h(S_2),
\end{aligned}
\tag{89}
$$

where (a) is due to the greedy manner of selecting subsequence $S_2^{\tau_2}$ from set $\mathcal{V} \setminus \mathcal{V}(S_1)$ and Lemma 6 (where we set $\alpha = 1$ and replace $\mathcal{V}$, $k$, and $i$ with $\mathcal{V} \setminus \mathcal{V}(S_1)$, $\tau_2$, and $\tau_2$, respectively), (b) holds because sequence $Z_2$ is a feasible solution to Problem $(P)$ for selecting a sequence of $\tau_2$ elements from $\mathcal{V} = \mathcal{V} \setminus \mathcal{V}(S_1)$, and (c) is from Eq. (83).

Therefore, we can characterize the value of $h(S_2)$ as follows (recall that $b' \triangleq \mu_1 \cdot \frac{k - \tau - \tau_2}{k - \tau_2}$):

$$
\begin{aligned}
h(S_2) &\geq \frac{e^{b'} - 1}{e^{b'} - a(\eta + \mu_2 - 1)} h(S^*(\mathcal{V} \setminus \mathcal{V}(S_1), k - \tau, 0)) \\
&\geq \frac{e^{b'} - 1}{e^{b'} - a(\eta + \mu_2 - 1)} g_\tau(S^*(\mathcal{V}, k, \tau)),
\end{aligned}
\tag{90}
$$

where the inequalities are from Lemma 7 (where we replace $\mathcal{V}$, $S$, $S_1$, $k$, $k'$, and $c$ with $\mathcal{V} \setminus \mathcal{V}(S_1)$, $S_2$, $S_2^{\tau_2}$, $k - \tau$, $k - \tau - \tau_2$, and $a(\eta + \mu_2 - 1)$, respectively) and Lemma 3, respectively.

Finally, we show that Eq. (87b) holds. We define two functions of $\eta$:

$$
l_1(\eta) = \frac{a \cdot (\eta + \mu_2 - 1)(e^{b'} - 1)}{e^{b'} - a \cdot (\eta + \mu_2 - 1)} \text{ and } l_2(\eta) = \frac{(1 - \eta)(e^{b'} - 1)}{e^{b'} - a \cdot (\eta + \mu_2 - 1)}.
$$

It can be verified that for $k > \tau + \tau_2$ and $\eta \in (0, 1]$, function $l_1(\eta)$ is monotonically increasing and function $l_2(\eta)$ is monotonically decreasing. Also, consider $\eta^* \triangleq \frac{1 + a - a \cdot \mu_2}{a + 1}$. We have $l_1(\eta^*) = l_2(\eta^*) = \frac{a\mu_2(e^{b'} - 1)}{(a+1)e^{b'} - a\mu_2}$. We consider two cases for $\eta$: $\eta \in [\eta^*, 1]$ and $\eta \in (0, \eta^*]$. For $\eta \in [\eta^*, 1]$, we have $\max\{l_1(\eta), l_2(\eta)\} \geq l_1(\eta) \geq l_1(\eta^*)$ as $l_1(\eta)$ is monotonically increasing; for $\eta \in (0, \eta^*]$, we have $\max\{l_1(\eta), l_2(\eta)\} \geq l_2(\eta) \geq l_2(\eta^*) = l_1(\eta^*)$ as $l_2(\eta)$ is monotonically decreasing. Therefore, for $\eta \in (0, 1]$, we have $\max\{l_1(\eta), l_2(\eta)\} \geq l_1(\eta^*) = \frac{a\mu_2(e^{b'} - 1)}{(a+1)e^{b'} - a\mu_2}$. This gives Eq. (87b) and completes the proof of Eq. (75). $\qquad \square$

## 8.11 Proof of Theorem 6

*Proof.* Suppose that function $h$ is forward-monotone, $\alpha$-backward-monotone, $\mu_1$-element-sequence-submodular, and $\mu_3$-general-sequence-submodular (Assumption 5). We use Lemma 3 presented in Appendix 8.3 and Lemma 6 presented in Appendix 8.8 to prove that Algorithm 2 achieves an approximation ratio of $\frac{\alpha^2 \mu_1 \mu_3(e^{\mu_1} - 1)}{(\mu_1 + \alpha\tau)e^{\mu_1}}$ in the case of $1 \leq \tau \leq k$, assuming the removal of an arbitrary subset of $\tau$ selected elements, which are not necessarily contiguous.

In Step 1 of Algorithm 2, sequence $S_1$ is selected by choosing $\tau$ elements with the highest individual values in a greedy manner, and we have $|S_1| = \tau$. In Step 2 of Algorithm 1, it is equivalent that sequence $S_2$ is selected by the SSG algorithm from set $\mathcal{V} \setminus \mathcal{V}(S_1)$, and we have $|S_2| = k - \tau$. Hence, the sequence selected by Algorithm 1 can be written as $S = S_1 \oplus S_2$. Recall that for any given sequence $S$, set $\mathcal{Z}_\tau(S)$ denotes the set of elements removed from sequence $S$ in the worst case (i.e., $\mathcal{Z}_\tau(S)$ is an optimal solution to Problem (11)). We define $\mathcal{Z}_\tau^1(S) \triangleq \mathcal{Z}_\tau(S) \cap \mathcal{V}(S_1)$ and $\mathcal{Z}_\tau^2(S) \triangleq \mathcal{Z}_\tau(S) \cap \mathcal{V}(S_2)$ as the set of elements removed from subsequences $S_1$ and $S_2$, respectively.

The proof of Theorem 6 follows a similar line of analysis as in the proof of Theorem 5. Specifically, we will also consider three cases: (I) $\mathcal{Z}_\tau^2(S) = \emptyset$, (II-a) $\mathcal{Z}_\tau^2(S) \neq \emptyset$ and $h(S_2) \leq h(S_2 - \mathcal{Z}_\tau^2(S))$, and (II-b) $\mathcal{Z}_\tau^2(S) \neq \emptyset$ and $h(S_2) > h(S_2 - \mathcal{Z}_\tau^2(S))$. The proofs of Case I and Case II-a are identical to those in Theorem 5. Therefore, we focus on Case II-b, which requires a different proof strategy.

We want to show the following bound that establishes the approximation ratio of Algorithm 2:

$$h(S - \mathcal{Z}_\tau(S)) \geq \frac{\alpha^2 \mu_1 \mu_3 (e^{\mu_1} - 1)}{(\mu_1 + \alpha\tau)e^{\mu_1}} g_\tau(S^*(\mathcal{V}, k, \tau)). \tag{91}$$

To begin with, we present a lower bound on $h(S_2)$, which will be used throughout the proof:

$$\begin{aligned} h(S_2) &\geq \alpha(1 - 1/e^{\mu_1})h(S^*(\mathcal{V} \setminus \mathcal{V}(S_1), k - \tau, 0)) \\ &\geq \alpha(1 - 1/e^{\mu_1})g_\tau(S^*(\mathcal{V}, k, \tau)), \end{aligned} \tag{92}$$

where the first inequality is from Lemma 6 (where we replace $\mathcal{V}$, $k$, and $i$ with $\mathcal{V} \setminus \mathcal{V}(S_1)$, $k - \tau$, and $k - \tau$, respectively) and the second inequality is from Lemma 3 (where we replace $\mathcal{V}'$ with $\mathcal{V}(S_1)$).

We now focus on Case II-b, where we have $\mathcal{Z}_\tau^2(S) \neq \emptyset$ and $h(S_2) > h(S_2 - \mathcal{Z}_\tau^2(S))$. Let $\eta \triangleq \frac{h(S_2) - h(S_2 - \mathcal{Z}_\tau^2(S))}{h(S_2)}$ denote the ratio of the loss of removing elements in $\mathcal{Z}_\tau^2(S)$ from sequence $S_2$ to the value of sequence $S_2$, and we have $\eta \in (0, 1]$. We first state the following:

$$h(S - \mathcal{Z}_\tau(S)) \geq \max\{\frac{\mu_1(\eta + \mu_3 - 1)}{\tau} \cdot h(S_2), \alpha(1 - \eta) \cdot h(S_2)\}, \tag{93a}$$

$$\max\left\{\frac{\mu_1(\eta + \mu_3 - 1)}{\tau}, \alpha(1 - \eta)\right\} \geq \frac{\alpha\mu_1\mu_3}{\mu_1 + \tau\alpha}. \tag{93b}$$

We will prove Eqs. (93a) and (93b) later; for now, we assume that they both hold. Then, we can obtain the following bound:

$$\begin{aligned} h(S - \mathcal{Z}_\tau(S)) &\geq \max\left\{\frac{\mu_1(\eta + \mu_3 - 1)}{\tau} \cdot h(S_2), \alpha(1 - \eta) \cdot h(S_2)\right\} \\ &\geq \max\left\{\frac{\mu_1(\eta + \mu_3 - 1)}{\tau}, \alpha(1 - \eta)\right\} \cdot \alpha(1 - 1/e^{\mu_1})g_\tau(S^*(\mathcal{V}, k, \tau)) \\ &\geq \frac{\alpha^2 \mu_1 \mu_3 (e^{\mu_1} - 1)}{(\mu_1 + \alpha\tau)e^{\mu_1}} g_\tau(S^*(\mathcal{V}, k, \tau)), \end{aligned} \tag{94}$$

where the three inequalities are from Eqs. (93a), (92), and (93b), respectively.

Now, we show that Eqs. (93a) and (93b) hold. We start by characterizing the value of elements in $\mathcal{Z}_\tau^2(S)$. Let $\tau_2 \triangleq |\mathcal{Z}_\tau^2(S)|$, and let the elements in $\mathcal{Z}_\tau^2(S)$ be denoted by $z_1, z_2, \ldots, z_{\tau_2}$ according to their order in sequence $S_2$. Then, we can rewrite $S_2$ as $S_2 = S_2^1 \oplus (z_1) \oplus S_2^2 \oplus (z_2) \oplus \cdots \oplus S_2^{\tau_2+1}$, where $S_2^i$ is the subsequence between elements $z_{i-1}$ and $z_i$, for $i = 1, 2, \ldots, \tau_2 + 1$, and both $z_0$ and $z_{\tau+1}$ are an empty sequence. Note that subsequence $S_2^i$ could be an empty sequence, for

$i = 1, 2, \ldots, \tau_2 + 1$. We characterize the value of elements in $\mathcal{Z}_\tau^2(S)$ in the following:

$$
\begin{aligned}
\eta \cdot h(S_2) &\overset{(a)}{=} h(S_2) - h(S_2 - \mathcal{Z}_\tau^2(S)) \\
&= h(S_2^1) + h((z_1)|S_2^1) + h(S_2^2|S_2^1 \oplus (z_1)) \\
&\quad + \cdots + h(S_2^{\tau_2+1}|S_2^1 \oplus (z_1) \oplus \cdots \oplus S_2^{\tau_2} \oplus (z_{\tau_2})) \\
&\quad - h(S_2^1) - h(S_2^2|S_2^1) - \cdots - h(S_2^{\tau_2+1}|S_2^1 \oplus \cdots \oplus S_2^{\tau_2}) \\
&= \sum_{i=1}^{\tau_2} h((z_i)|S_2^1 \oplus (z_1) \oplus \cdots \oplus (z_{i-1}) \oplus S_2^i) + h(S_2^2|S_2^1 \oplus (z_1)) - h(S_2^2|S_2^1) \\
&\quad + \cdots + h(S_2^{\tau_2+1}|S_2^1 \oplus (z_1) \oplus \cdots \oplus S_2^{\tau_2} \oplus (z_{\tau_2})) - h(S_2^{\tau_2+1}|S_2^1 \oplus \cdots \oplus S_2^{\tau_2}) \\
&\overset{(b)}{\leq} \sum_{i=1}^{\tau_2} h((z_i)|S_2^1 \oplus (z_1) \oplus \cdots \oplus (z_{i-1}) \oplus S_2^i) \\
&\quad + h(S_2^2|S_2^1 \oplus (z_1)) - \mu_3 \cdot h(S_2^2|S_2^1 \oplus (z_1)) \\
&\quad + \cdots + h(S_2^{\tau_2+1}|S_2^1 \oplus (z_1) \oplus \cdots \oplus S_2^{\tau_2} \oplus (z_{\tau_2})) \\
&\quad - \mu_3 \cdot h(S_2^{\tau_2+1}|S_2^1 \oplus (z_1) \oplus \cdots \oplus S_2^{\tau_2} \oplus (z_{\tau_2})) \\
&= \sum_{i=1}^{\tau_2} h((z_i)|S_2^1 \oplus (z_1) \oplus \cdots \oplus (z_{i-1}) \oplus S_2^i) + (1 - \mu_3) \cdot h(S_2^2|S_2^1 \oplus (z_1)) \\
&\quad + \cdots + (1 - \mu_3) \cdot h(S_2^{\tau_2+1}|S_2^1 \oplus (z_1) \oplus \cdots \oplus S_2^{\tau_2} \oplus (z_{\tau_2})) \\
&= \sum_{i=1}^{\tau_2} h((z_i)|S_2^1 \oplus (z_1) \oplus \cdots \oplus (z_{i-1}) \oplus S_2^i) \\
&\quad + (1 - \mu_3)(h(S_2^2|S_2^1 \oplus (z_1)) + \cdots + h(S_2^{\tau_2+1}|S_2^1 \oplus (z_1) \oplus \cdots \oplus S_2^{\tau_2} \oplus (z_{\tau_2}))) \\
&\overset{(c)}{\leq} \sum_{i=1}^{\tau_2} h((z_i)|S_2^1 \oplus (z_1) \oplus \cdots \oplus (z_{i-1}) \oplus S_2^i) + (1 - \mu_3) \cdot h(S_2) \\
&\overset{(d)}{\leq} \sum_{i=1}^{\tau_2} h((z_i))/\mu_1 + (1 - \mu_3) \cdot h(S_2),
\end{aligned}
$$

(95)

where (a) is from the definition of $\eta$, (b) is due to function $h$ being $\mu_3$-general-sequence-submodular, (c) holds because $\mu_3 \leq 1$ and $S_2 = S_2^1 \oplus (z_1) \oplus S_2^2 \oplus (z_2) \oplus \cdots \oplus S_2^{\tau_2+1}$, and (d) is due to function $h$ being $\mu_1$-element-sequence-submodular. From Eq. (95), we have the following:

$$
\sum_{i=1}^{\tau_2} h((z_i)) \geq \mu_1(\eta + \mu_3 - 1)h(S_2). \tag{96}
$$

To prove Eq. (93a), we decompose it into two parts: (i) $h(S - \mathcal{Z}_\tau(S)) \geq \frac{\mu_1(\eta+\mu_3-1)}{\tau} \cdot h(S_2)$ and (ii) $h(S - \mathcal{Z}_\tau(S)) \geq \alpha(1 - \eta) \cdot h(S_2)$.

Let $v'$ denote the first element in $S_1 - \mathcal{Z}_\tau^1(S)$. Then, Part (i) can be shown through the following:

$$
h(S - \mathcal{Z}_\tau(S)) \overset{(a)}{\geq} h((v')) \overset{(b)}{\geq} \frac{1}{\tau_2} \sum_{i=1}^{\tau_2} h((z_i)) \overset{(c)}{\geq} \frac{\mu_1(\eta+\mu_3-1)}{\tau_2} \cdot h(S_2) \overset{(d)}{\geq} \frac{\mu_1(\eta+\mu_3-1)}{\tau} \cdot h(S_2),
$$

where (a) is due to the forward monotonicity of function $h$, (b) is due to $h(v') \geq h(z_i)$ for any $i = 1, 2, \ldots, \tau_2$, (c) is from Eq. (96), and (d) is due to $\tau_2 \leq \tau$.

Part (ii) can be shown through the following:

$$
h(S - \mathcal{Z}_\tau(S)) = h((S_1 - \mathcal{Z}_\tau^1(S)) \oplus (S_2 - \mathcal{Z}_\tau^2(S))) \overset{(a)}{\geq} \alpha \cdot h(S_2 - \mathcal{Z}_\tau^2(S)) \overset{(b)}{=} \alpha(1 - \eta) \cdot h(S_2),
$$

where (a) is from the $\alpha$-backward monotonicity of function $h$ and (b) is from the definition of $\eta$.

Eq. (93b) holds trivially for any $\eta \in (0, 1]$ by setting $\frac{\mu_1(\eta+\mu_3-1)}{\tau}$ and $\alpha(1 - \eta)$ to be equal, solving for $\eta$, and plugging it back. This completes the proof. $\qquad\square$