[Reviews · NeurIPS 2020]

Review 1

Summary and Contributions: This paper considers the problem of robust sequence submodular optimization. In this setting, the function value depends on the order in which the items are selected, and must be robust to adversarial deletion of elements from the sequence. Algorithms under a variety of notions of (approximate) sequence submodularity are formulated.

Strengths: + Algorithms for robust sequence submodular optimization are formulated and worst-case guarantees are proven. + The authors distinguish between different versions of sequence submodularity (element-wise or more general) and analyze their algorithms in multiple of these settings. Apparently these versions of sequence submodularity were previously thought to be the same.

Weaknesses: - It is unclear how difficult the generalization from algorithms for robust submodularity to robust sequence submodularity is. The authors do not explain this relationship, and it feels like a routine generalization, at least in parts. - Did not find any of the applications particularly well motivated from ML perspective. In particular, for movie recommendation, most movies are stand-alone, rather than dependent on a sequence. For movies that form explicit sequences, it seems unlikely that the user would adversarially skip some of the sequence. - Lack of empirical evaluation of the proposed approach. - The algorithms are very simple greedy approaches. The simplicity of the algorithms, combined with the lack of context on algorithms in the robust set case, is a drawback.

Correctness: Appears to be. Did not check completely.

Clarity: The paper is readable, although parts (such as the Intro.) feel overly verbose. The intro is 3 pages long. Large blocks of text are italicized.

Relation to Prior Work: Insufficient, see comments.

Reproducibility: Yes

Additional Feedback: WIthout information on how the algorithms for robust sequence submodularity differ from previous work on algorithms for robust set submodularity, it is difficult to judge the novelty of the submission. Do the ratios obtained here match the robust set case? Are the algorithms similar, with the substitution of a sequence submodular function for a set submodular one? These algorithms (for robust sets) would also be good to compare with empirically if the authors could find a suitable application. ===== edit after rebuttal ===== After reading the authors' feedback and other reviews, my opinion of the paper hasn't changed. I think a substantially improved version of this paper, with better motivation, better context, perhaps an empirical evaluation that demonstrates differences between robust sequence and set approaches, could be accepted. But, these are too many changes for the camera-ready version and in my opinion would require another round of review. === second edit After further discussion, I looked over the submission and response again, and there were a couple of points I didn't properly appreciate: + the element-sequence-submodular vs sequence-submodular settings, where the authors provide ratios in both settings + Given how much of the analysis depends on properties unique to the sequence setting, such as forward-monotonicity, backward-monotonicity, etc., I suppose any analogous algorithms for sets would have much simpler analysis (even if the underlying ideas were similar). Therefore, I will revise my score upwards by one point.


Review 2

Summary and Contributions: This paper studies the sequence submodular maximization problem. It generalizes the submodular set functions to the sequence functions where the order of adding elements is important and determines the function. They consider the robustness of sequence submodular functions against the removal of a subset of elements in the selected sequence. The definition of the sequence submodular function is borrowed from [7]. They design two algorithms for this problem. === After author response === I have read the author's feedback. The algorithmic contribution of this work is two greedy like algorithms. In my opinion, these two algorithms are the first baselines that one can propose for this problem. Given the lack of substantial algorithmic contribution, experimental evaluations could strengthen the paper by showing the efficiency of these simple baseline algorithms in practice. On the other hand, there are several algorithms for robust "set" submodular maximization. I believe it is necessary to discuss the practical advantage of the proposed approach over them for the "sequence" submodular maximization problem. As a result, I would keep my score unchanged.

Strengths: 1. It studies an important problem with potential real-world applications. 2. The paper is well written and easy to read. 3. The proofs are correct. 4. It could have real-world applications.

Weaknesses: My main concern regarding this paper is the strength of the theoretical results. The two proposed algorithms are simple and I believe the guarantees are in line with what I would expect from such algorithms. Also, Algorithm 2 provides a constant factor approximation for the case of removing an only constant number of elements. Next, I provide a few other issues that it would be great to be addressed by the authors: 1. The assumption of contiguous removal of elements seems unrealistic as it imposes an artificial restriction on the set of deleted elements. 2. [13] and [14] provide a more sophisticated algorithm for the deletion of robust submodular maximization. It is beneficial to extend those ideas to the case of sequence submodular functions. 3. As this is a problem with many participial applications, evaluating the performance of the algorithms is necessary. 4. The reference to [MFKK] which is highly related to this problem is missing. Also, [6,12] and [MFKK] are studying the sequence submodular maximization with a slightly different definition. I would like to see the impact of the proposed algorithms on those problems. [MFKK] Mitrovic, Marko, Moran Feldman, Andreas Krause, and Amin Karbasi. "Submodularity on Hypergraphs: From Sets to Sequences." In International Conference on Artificial Intelligence and Statistics, pp. 1177-1184. 2018.

Correctness: As far as I checked, the theoretical results are correct. The paper does not provide any experimental evaluaitons.

Clarity: Yes, the paper is well written and easy to follow.

Relation to Prior Work: One relevant refrence is missing. Although the other related papers are cited, the relation with those algorithm are missing. [6,12] and [MFKK] are studying the sequence submodular maximization with a slightly different definition. I would like to see the impact of the proposed algorithms on those problems. Also, it is not discussed why the ides of [13] and [14] are not applicable to the case of sequence submodular functions.

Reproducibility: No

Additional Feedback: 1. Provide algorithms with stronger theoretical guarantees. 2. Evaluate the performance of the algorithms empirically. 3. Explain the relation with the previous works on both sequence submodular maximization and deletion robust submodular maximization. Elaborate the differences and explain why those previous techniques for deletion robust problem are not working here.


Review 3

Summary and Contributions: This paper considers a generalization of submodularity called sequence submodularity. This is intended to provide a framework in which valuations depend on the orderings of items. For example, a user's value for some goods might depend on the order in which the goods are presented. It is not clear precisely how to define sequence submodularity, and there are quite a few subtleties in the definitions. The authors discuss several definitions and the relations among them. They show have a counterexample show that a relation that was presumed to be true in prior work are in fact not true. Cardinality-constrained maximization for monotone submodular functions was previously known to generalize to sequence-submodular functions. The main focus of this work is on extend such results for *robust* optimization: that is, settings in which an adversary can select at most \tau elements for removal from the chosen set. Due to the sequential ordering of the items, the authors consider settings in which the removed items must be sequential or could be arbitrary. The authors present approximation algorithms with provable approximation ratios as a function of \tau (the robustness) and k (the cardinality). They also generalize these results to settings in which submodularity holds only approximately.

Strengths: I found this setting to be quite interesting. The algorithms and analyses seemed novel to me. I think that these results would appeal to a reasonable fraction of the NeurIPS community.

Weaknesses: The approximation ratios are not very clean, but of course this could be inherent in the problem. I acknowledge the authors' rebuttal that approximation ratios independent of tau and k are present. I did not consider the detailed form of the approximation ratios to be a significant drawback of the paper.

Correctness: As far as I could tell the claims are correct.

Clarity: The paper was clearly written.

Relation to Prior Work: Yes.

Reproducibility: Yes

Additional Feedback:


Review 4

Summary and Contributions: This paper extends the sequence submodular maximization problem (initialized in [7]) to the robust setting where some terms in the sequence are allowed to be removed. Several algorithms have been proposed and both approximation ratios and computational complexity have been established. Those results can be considered as interesting contributions to the submodular optimization theory.

Strengths: - This paper is well-written and all materials are organized in a very reasonable way, I enjoy reading this paper. In general, this paper is mathematically elegant. - The technical part of this paper is solid. Proving the approximation ratios of the proposed algorithms is quite non-trivial. - Submodular optimization is generally important to many machine learning problems. The variant studied in this paper may also be useful in some settings.

Weaknesses: - I am not convinced by the motivating examples given in the paper. It is still unclear to me how those problems (e.g., sensor replacement & movie recommendation) can be modeled as robust sequence submodular optimization problems and why we should model it in that way. I am concerning the practical value of this paper. - The proposed methods have not been validated by experiments. In fact, I am curious if there are some benchmarks that can be modeled as robust sequence submodular problems. - Some of the bounds are not very practical. E.g. the bound in Theorem 3 is almost useless for large \tau.

Correctness: I didn’t check all the detailed proofs, but the overall claims & theorems look reasonable to me.

Clarity: A well written paper.

Relation to Prior Work: Yes

Reproducibility: Yes

Additional Feedback: - Can you give a few concrete functions that satisfy the required properties? - I am not convinced by the motivating examples. It is unclear to me how those examples can be modelled as the proposed robust sequence submodular maximization problems. Can you add some explanation? - Footnote 3 is splitted into two pages, better to put it in the same page. - Line 224, “two reasons” -> “for two reasons” - The bound in Theorem is quite non-trivial, it will be helpful to create a table/figure to illustrate how the value of this bound changes when we change k and \tau. - Theorem 3 is very weak for large \tau, the approximation ratio is almost useless. - Ideally, the proposed methods should be verified by some experiments. Experiments can also be used to justify that the proposed methods do find some real-world applications. ======= After reading the rebuttal letter ================ A theory paper without empirical evaluation is okay to me, but I think my biggest concern -- lack of strong motivations/applications -- has not yet been well addressed in the rebuttal letter. I would keep my decision and still vote for a weak accept.

[Author Response · NeurIPS 2020]

[**Reviewer 1**] **1.1 Robust sequence submodular vs. robust set submodular.** The main differences are two-fold. **(i)** From the algorithmic perspective, while there are some similarities in the designed algorithms, Algorithm 1 is designed specifically for the special case of the removal of contiguous elements and achieves a constant approximation ratio for any value of $\tau$ and $k$. This is different from the algorithms in [13][14], which achieve constant approximation ratios only when $\tau$ is very small compared to $k$. **(ii)** The analysis with respect to sequence functions is more challenging, which mainly stems from the new properties (that are irrelevant for set functions), such as multiple (nonequivalent) definitions of the diminishing returns property, two forms of monotonicity, and the impact of the ordering itself. Any attempt of adapting algorithms from robust set submodularity needs to carefully consider the aforementioned subtle yet critical differences while establishing approximation guarantees. The algorithms for robust set submodularity cannot be directly applied to sequence functions, as converting a set into a sequence could result in an arbitrarily bad performance. **1.2 Applications.** The order is important in many cases, e.g., when the recommended videos are part of a movie series or a TV show. In fact, movie recommendation and TV show recommendation have been modeled as sequence functions in [12] and [6], respectively. As noted in the motivating example in [12], if the model determines that the user might be interested in *The Lord of the Rings* series, then recommending *The Return of the King* first and *The Fellowship of the Ring* last could make the user unsatisfied with an otherwise excellent recommendation. Moreover, the user may not watch all the recommended videos possibly because the user has already watched some of them or does not like them (e.g., due to low ratings and/or unfavorable reviews). While the objective functions in some applications may not always be sequence submodular, in Section 4, we introduce generalized formulations that account for *approximate sequence submodularity* and *approximate backward monotonicity* and leverage them to prove approximation results of the proposed algorithms under weaker assumptions, which we believe hold for a wider range of applications. **1.3 Empirical evaluation.** We agree that empirical evaluation is important. However, as a very first study on this new problem, we choose to focus on fundamentals and rigorous analyses, such as designing and theoretically analyzing algorithms, introducing and clarifying subtle yet critical properties of sequence functions, proving approximation guarantees for important variants of the problem, and providing useful insights for further studies. This is in line with other relevant work that investigates new problems and focuses on theoretical studies, such as robust set submodular maximization [13] and sequence submodular maximization [9]. We believe that this work serves as an important first step towards the design and analysis of efficient algorithms for robust sequence submodular maximization, which can be further explored through empirical evaluations for specific applications. **1.4 Simple algorithms.** While our greedy algorithms are simple, the theoretical analysis is more challenging, and the presented approximation guarantees are highly nontrivial. Please also see "1.1 Robust sequence submodular vs. robust set submodular."

[**Reviewer 2**] **2.1 Theoretical results.** Algorithm 2 allows the removal of arbitrary $\tau$ elements, and thus, it is not surprising that achieving a stronger approximation guarantee (than that in Theorem 3) becomes more challenging if not impossible. On the other hand, Algorithm 1 achieves a constant approximation ratio for any value of $\tau$ in the special case of the removal of contiguous elements. Note that even for robust set submodularity, to the best of our knowledge, the developed algorithms achieve a constant approximation ratio only when $\tau$ is very small compared to $k$ [13][14]. **2.2 Contiguous removal of elements.** The assumption of the removal of contiguous elements can model a spatial relationship such as sensors in close proximity or a temporal relationship such as consecutive episodes of a TV show. We exploit the properties of such a special case to design Algorithm 1, which achieves a constant approximation ratio for any value of $\tau$. Note that there is no such special case in robust (set) submodular maximization. **2.3 Extension of [13][14].** It is unclear whether the algorithms in [13][14] can be properly extended to our problem, and even if so, it is more likely that establishing their approximation guarantees would require a more sophisticated analysis, which calls for more in-depth investigations. We believe that our work serves as an important first step towards developing efficient algorithms for robust sequence submodular maximization. Note that the analysis of our simple greedy algorithms is already very sophisticated. Please also see "1.1 Robust sequence submodular vs. robust set submodular." **2.4 Evaluation.** Please see "1.3 Empirical evaluation." **2.5 References [6][12][MFKK].** While these references assume that the sequential relationship among elements is encoded as a directed acyclic graph, we consider a general setting without such structures. It would indeed be interesting to explore our algorithms when the sequential relationship is encoded in a specific graphical form. We will elaborate on such discussions in the revised version.

[**Reviewer 3**] **3.1 Approximation ratios.** The approximation ratio in Theorem 3 is pretty clean. The approximation ratio in Theorems 1 and 2 is the maximum of two terms (see supplementary files): the first is a constant; the second depends on $\tau$ and $k$. Thus, it is lower bounded by a (clean) constant *independent* of $\tau$ and $k$. We include the second term as it leads to a better overall approximation ratio for a larger $k$. The additional parameters ($\alpha$ and $\mu$'s) render the approximation ratios in Theorems 4-6 somewhat complex, but they are necessary for the generalized formulations.

[**Reviewer 4**] **4.1 Motivating examples.** Please see "1.2 Applications." **4.2 Experiments.** Please see "1.3 Empirical evaluation." **4.3 Bounds.** Please see "2.1 Theoretical results." **4.4 Concrete functions.** Please refer to [5] for some concrete sequence submodular functions, such as the one in Eq. (10) of [5], which is the expected fraction of accomplished subtasks. **4.5 Bounds plot.** Thanks for the suggestion. We will add such figures in the revised version.

[Meta-Review · NeurIPS 2020]

We had many rounds of discussion about the contributions of this paper. The authors need to better relate their paper to the huge body of robust optimization. Also, as reviewers mentioned, some references were missing. However, at the end, we came to the conclusion that the theoretical contribution of the paper is sold and merits acceptance. So I highly suggest that the authors, and we trust that you will, take reviewer's comments seriously into account to improve the quality of the final version.